

# The density–salinity relation of standard seawater

Hannes Schmidt[1], Steffen Seitz[1], Egon Hassel[2], Henning Wolf [1]

[1]Physikalisch-Technische Bundesanstalt, Braunschweig, 38116, Germany
[2]Lehrstuhl für Technische Thermodynamik, Universität Rostock, Rostock, 18051, Germany

*Correspondence to*: H. Wolf (henning.wolf@ptb.de)

**Abstract.** The determination of salinity by means of electrical conductivity relies on a constant salt composition in the North Atlantic Ocean, as standard seawater, which is required for salinometer calibration, is produced therefrom. In order to verify the long-term constant composition of standard seawater, it was proposed to perform density measurements on standard seawater, as the seawater density is sensitive to all salt components. Thus, density measurements can detect any change in the

composition of seawater. A conversion of the density values to salinity can be performed by means of a density–salinity relation. To use such a relation with a target uncertainty in salinity comparable to that in salinity obtained from conductivity measurements, a density measurement at an uncertainty level of 2 ppm is mandatory. In this article, a new density–salinity relation is presented based on such accurate density measurements. The underlying substitution measurement method is described, density corrections for uniform isotopic and chemical compositions are reported, and the density–salinity relation

is presented. The comparison of densities calculated using the new relation with those calculated using the present reference equation of state TEOS-10 suggests that the density accuracy of TEOS-10 and that of some of its underlying density measurements are overestimated. The new density–salinity relation may be used to verify the constant composition of standard seawater by means of routine density measurements.

## 1 Introduction

For almost 40 years, the salinity[3] of seawater has been indirectly determined by means of electrical conductivity. Since the absolute conductivity cannot be measured as accurately as required for precise salinity measurements (Seitz et al., 2010), the conductivity has been measured relative to that of standard seawater[4]; the conversion to salinity is carried out by means of the (relative) conductivity–salinity relation PSS-78 (JPOTS, 1981a and b). In practice, this is achieved by calibrating salinometers and conductivity-temperature-depth devices using standard seawater, which is diluted to obtain the conductivity of the

potassium chloride standard (Culkin, 1986; Bacon et al., 2007) used as conductivity reference. An unconditional prerequisite for the comparability of salinity measurements over long periods is, therefore, that standard seawater has a stable chemical composition. Unfortunately, this cannot be guaranteed, as standard seawater is of natural origin.

Recently, the long-term comparability of salinity measurement results was discussed, with two main deficiencies being elaborated (Pawlowicz et al., 2016): a lack of traceability to a long-term stable and ubiquitous reference like the International

System of Units SI and chemical composition variabilities in standard seawater. These variabilities are likely to increase in the

---

[3]    ‘Salinity’ refers strictly to practical salinity unless there is an exact specification.
[4]    Standard seawater recognized by the International Association for the Physical Sciences of the Oceans (IAPSO) prepared from seawater of the Northern Atlantic Ocean.



coming decades, due especially to the absorption of carbon dioxide into the ocean resulting from accumulation in the atmosphere (Millero, 2007). Both of these deficiencies entail a risk of inconsistent long-term salinity values. In order to remedy the deficiencies, Seitz et al. (2011) proposed to perform routine density measurements on standard seawater. In practice, this would be achieved by determining the salinity of a standard seawater batch not only by a conductivity measurement but also

by a density measurement; the conversion to salinity is carried out in this second approach by means of a density–salinity relation. Since the salinity obtained from density is sensitive to all components of the standard seawater, a change in its composition would lead to an inconsistency of the "density salinity" and the "KCl salinity".

In order to obtain a reliable statement about the consistency of the density salinity and the KCl salinity, they have to be compared against the background of their uncertainties. The reproducibility of the KCl salinity is 0.0004 (Bacon et al., 2007).

However, this reproducibility is only valid for the time of preparation (Seitz et al., 2010), as, during storage, glass container material dissolves in the seawater, which is mainly silicon (e.g. Poisson, 1978; Higgs and Ridout, 2011; Uchida et al., 2011). Note that the uncertainty in the "conductivity salinity" obtained by means of a salinometer is at least 0.0022 (Le Menn, 2011), and requires freshly prepared standard seawater for calibration. The corresponding values in terms of density are 0.3 g m$^{-3}$ (for 0.0004) und 1.8 g m$^{-3}$ (for 0.0022). The present reference equation of state TEOS-10 (IOC et al., 2010) summarizes the most

accurate density measurements obtained from standard seawater conducted by Millero et al. (1976) and by Poisson et al. (1980). TEOS-10, which implicitly contains a density–salinity relation of standard seawater, predicts the density with an estimated uncertainty of at least 8 g m$^{-3}$ (Feistel, 2008), which is significantly higher than 1.8 g m$^{-3}$, but reflecting the measurement uncertainty in seawater density at that time.

In this article, a new density–salinity relation is presented, whereby the salinity can be determined by means of density

measurement with an accuracy of up to 0.003 for salinities up to 35, temperatures between 5 °C and 35 °C and atmospheric pressure, which is similar to the accuracy achieved by salinometers. The density was determined by using the substitution method developed by Schmidt et al. (2016). Because the water-isotopic and salt-chemical compositions, as well as the air saturation, of the seawater samples changed during preparation, storage and measurement, corrections were applied to specify the seawater density for uniform conditions; these corrections are of the same order of magnitude as the measurement

uncertainty and are therefore essential for high accuracy. The corrected density values were used to develop a density–salinity relation. The comparison of densities calculated by means of the new relation with those calculated by means of TEOS-10 suggests that TEOS-10 predicts densities significantly too high by up to 15 g m$^{-3}$. The deviations increase systematically with salinity. A plausible explanation was found in the design of the flotation densimeter (Millero, 1967) that Millero et al. (1976) used for their measurements obtained from standard seawater.

The new density–salinity relation may be used to reliably verify the constant composition of standard seawater by means of routine density measurements. On the one hand, the determination of salinity by means of conductivity is retroactively ensured in case of consistency; on the other hand, in case of inconsistency, a need for action is demonstrated.

## 2 Density measurements

Determining salinity by means of conductivity measurement is supported by the relations of PSS-78. In order to develop the

density–salinity relation in such a way that it is consistent with PSS-78, the density measurements have to be obtained from seawater whose salinity determination is consistent with the salinity determination of the seawater used to develop the PSS-78 relations. In addition to the consistency of the salinity determination, the accuracy of the density measurement is decisive. The more accurate the density measurement is, the more accurately the salinity can be determined (by means of the density–



salinity relation). In order to achieve an accuracy in the density salinity that is equal to that in the conductivity salinity, a density uncertainty of 2 g m$^{-3}$ is required. To this end, substitution measurement with a vibrating-tube densimeter relative to a water reference had been proposed (Wolf, 2008) before a substitution method specifically for seawater was developed and validated (Schmidt et al., 2016).

In this section, the preparation of the seawater measured and the determination of its salinity are described. The consistency of the salinities determined in the present, which were used to develop the density–salinity relation, with the salinities determined in 1978, which were used to develop PSS-78, is discussed. The substitution method and the apparatus used for the density measurement are briefly outlined, as they have already been described in detail by Schmidt et al. The uncertainty in density is discussed with regard to the uncertainty in salinity obtained from a density measurement and the subsequent

calculation by means of the density–salinity relation.

### 2.1 Substitution method

In a substitution method, a sample (= seawater) with an unknown density and a similar, well known reference (= water) are measured (ideally, at the same time) using the same measurement device (= densimeter). Deviations in the measurement results caused, for example, by a drift or a temperature deviation can be corrected, as they cause similar effects on seawater and on

water. As a result, the measured densities of seawater and water have similar deviations from their true value. The difference equation for calculation of the corrected density from the measurements obtained from seawater and water is:

$$\rho_{\text{subs}}^{\text{SW}} - \rho_{\text{ref}}^{\text{H}_2\text{O}} = \rho_{\text{mes}}^{\text{SW}} - \rho_{\text{mes}}^{\text{H}_2\text{O}}, \tag{1}$$

where $\rho_{\text{mes}}^{\text{SW}}$ and $\rho_{\text{mes}}^{\text{H}_2\text{O}}$ are the measured seawater and water densities, and $\rho_{\text{subs}}^{\text{SW}}$ and $\rho_{\text{ref}}^{\text{H}_2\text{O}}$ are the corrected seawater (substitution) density and the well-known water reference density. If the absolute seawater density is determined from a

substitution measurement (by calculating $\rho_{\text{mes}}^{\text{SW}} - \rho_{\text{mes}}^{\text{H}_2\text{O}} + \rho_{\text{ref}}^{\text{H}_2\text{O}}$), the result includes the uncertainty in the water reference density. By contrast, if the seawater density relative to water is determined (by calculating $\rho_{\text{mes}}^{\text{SW}} - \rho_{\text{mes}}^{\text{H}_2\text{O}}$), the reference uncertainty is not included.

The water reference density was calculated using the equation of state developed by Wagner and Pruß (2002). A description of the calculation is given in Appendix A. The reference density uncertainty is 1 g m$^{-3}$ for atmospheric pressure, 10 g m$^{-3}$ for

pressures up to 10 MPa, and 30 g m$^{-3}$ for up to 100 MPa. The uncertainty in a corrected seawater density resulting from a substitution measurement mainly depends on the uncertainty in the water reference density, but also on the similarity of seawater and water in terms of their relevant thermophysical properties, as well as on the stability and linear characteristics of the densimeter used.

### 2.2 Materials

#### 2.2.1 Reference water

The water used as the reference liquid in the substitution measurements was prepared using tap water from Braunschweig, Germany. It was purified using a reverse osmosis module, an ion exchanger, and a 0.2 μm-filter. Its purity was checked by measuring the water conductivity at the outlet of the filter; the conductivity at 20 °C to 25 °C was always lower than 0.1 μS cm$^{-1}$. The water was degassed by boiling it for half an hour under minimum power. Immediately afterwards, it was

poured into borosilicate vessels that were sealed in a hot state. This water was used for measurements over the course of one week. The reference water-air saturation was 20 % with an uncertainty of 10 %. The isotopic abundances of deuterium and of



oxygen-18 against Vienna Standard Mean Ocean Water were −59 ‰ and −8.5 ‰, respectively. Details on the determination of the air saturation and isotopic composition were given by Schmidt et al. (2016).

### 2.2.2 Seawater

All seawater samples were obtained from Ocean Scientific International Ltd. (OSIL), Havant, UK, which also determined the

salinity values. Samples with salinities of 10, 30, and 35 were taken from batches 10L13, 30L15, and P153, respectively. Additionally, diluted seawater with salinities of 5, 15, 20, and 25 was studied. These seawater batches were prepared using the same procedure as that used for the standard batches with salinities 10 and 30: First, a large amount of natural seawater (as used for the preparation of standard seawater) was diluted with water until its salinity was approximately equal to the target salinity. The raw salinity was determined using a modified 8400B Autosal salinometer (Bacon et al., 2007). Then, for

calibration, a set of five samples per salinity was obtained by means of weight dilution of standard seawater (from batch P154 with a salinity of 34.9962). The balance used had a readability of 0.1 mg and was calibrated using weight standards traceable to the National Physical Laboratory, Teddington, UK (B. Childs, personal communication, 2017). The salinity was again determined by the Autosal salinometer, on the one hand, and by means of the weights of the standard seawater and the water used for dilution on the other hand. The deviations found between the salinometer salinities and the weight-calculated salinities

were used as calibration offsets for the raw salinities of the diluted seawater.

The salinity homogeneity and calibration measurements yielded the values and corresponding standard deviations given in Table 1. The uncertainty in the salinity of standard seawater was adopted from Bacon et al. (2007). The uncertainty in the salinity of diluted seawater includes the standard deviations of homogeneity and calibration measurements, as well as the uncertainty in the salinity of standard seawater. The systematic uncertainty contributions of weighing and refilling are

negligible compared to the standard deviations. The uncertainty in the salinity of dilute samples is 0.0006, which corresponds to a density uncertainty of 0.5 g m$^{-3}$.

### 2.3 Apparatus

Vibrating-tube densimeters (VTDs) were used for density measurements performed using the substitution method. The core of such a densimeter is a U-shaped tube that is fixed in place on both ends. This tube is filled with the liquid to be measured

and then forced to oscillate; the resulting oscillation period is a measure of the density of the liquid. Since the vibrating tube can be easily accessed from the outside, liquids can be filled in and changed quickly. This feature, together with short-term stability, is necessary for the application of the substitution method. Since the seawater sample and water reference cannot be measured simultaneously, stability is important for the duration of the alternating measurements. Under these conditions, the drift of the densimeter can be quantified using the deviations from the reference density (of water) to correct the sample density

(of seawater).

The set-up used for the density measurements at atmospheric pressure is outlined in Fig. 1a. It comprises a fully automated filling system, a VTD and a peristaltic pump. The filling system was created specifically for small filling volumes in order to allow more repetitions in the substitution measurements from a limited sample amount. To this end, a sequence of humid air bubbles is used to rinse the previous liquid out of the measuring cell. The bubbles of humid air are inserted into the sample

filling tubes using the V2 and V3 valves in addition to the V1 valve to switch between the seawater and the reference water. The VTD used for the measurements is a DMA 5000M (Anton Paar GmbH, Graz, Austria). The peristaltic pump used to move the liquids is installed behind the VTD to avoid any interaction of the peristaltic tube material with the seawater or the water before the measurement.





The set-up used for density measurements at high pressures is illustrated in Fig. 1b. It uses an equal filling system to fill the water and seawater like the set-up used for atmospheric pressure. In addition to the filling system, the VTD, and the peristaltic pump, a pressurization part is installed between the VTD and the peristaltic pump. In this part, wherein the pressure is generated and measured, is a syringe pump filled with oil to prevent corrosion of the pressure sensors. The oil transmits the pressure

generated in the syringe pump directly to the water without using a pressure transmitter. A long tube is installed between both parts (VTD and pressurization part) to avoid diffusion of oil into the measurement cell of the VTD. Two pressure sensors (P1 up to 14 MPa and P2 up to 70 MPa) are used to increase the accuracy of the pressure measurement. The offsets of these sensors at atmospheric pressure are corrected by the values gained with the atmospheric pressure manometer before each measurement. The VTD used for the measurements at high pressures is a DMA HP (Anton Paar GmbH, Graz, Austria).

The substitution measurements at atmospheric pressure were performed at a constant temperature. The water and seawater were filled and measured in alternation. The water densities measured were thus compared with the reference; the deviations found were used to correct the seawater measurements.

The procedure for high pressures is similar to that used for atmospheric pressure; however, the liquid is not changed during a high pressure run at a constant temperature. Instead, the liquid is changed after decreasing the pressure back to atmospheric

conditions.

### 2.4 Substitution densities

The seawater density was measured in the temperature range of 5 °C to 35 °C. The densities were corrected to integer temperatures in °C and either to 101325 Pa or to integer pressures in bar, if the substitution density was determined for high pressures. The resulting absolute seawater densities have uncertainties of 2 g m$^{-3}$ for atmospheric pressure, 14 g m$^{-3}$ for

pressures up to 10 MPa, and 34 g m$^{-3}$ for pressures up to 65 MPa. If stated relative to water, the seawater densities for high pressures have significantly smaller uncertainties, as they do not include the water reference uncertainty. In this case, the relative density uncertainties are 5 g m$^{-3}$ for pressures up to 10 MPa and 6 g m$^{-3}$ for pressures up to 65 MPa.

Since the salinity uncertainty, which is 0.5 g m$^{-3}$ in terms of density, is significant compared to the density measurement uncertainty, it has to be considered in the development of the density–salinity relation. This had already been done at this point

by adding the salinity uncertainty to the density measurement uncertainty.

### 2.5 Comparability of salinity

For determining salinity by means of conductivity, the PSS-78 relations were developed based on five datasets[5], which comprise conductivity measurements obtained from potassium chloride solutions and from standard seawater solutions with salinities of 2 to 42. Standard seawater obtained from batch P79 was used in order to define the reference point at salinity 35.

To this end, the mass fraction of the potassium chloride solution which has the same conductivity as standard seawater (with salinity 35) was determined. These measurements were reported by Culkin and Smith (1980), Dauphinee et al. (1980a) and Poisson (1980a). Standard seawater obtained from the batches P73, P75, and P79 was used to determine the conductivity of (diluted and concentrated standard seawater with) salinities $\neq 35$ relative to (seawater with) a salinity of 35. These

---

[5] Note that all publications cited here were also reprinted together (JPOTS, 1981b).





measurements were reported by Bradshaw and Schleicher (1980), Dauphinee et al. (1980b) and Poisson (1980b). For weighing, very precise balances were used, e.g. a Mettler M5 GD with a precision of 1 µg for the potassium chloride or a Mettler B5 C1000 with a precision of 0.1 mg for the solutions. The five datasets were used by Perkin and Lewis (1980) to find the coefficients of empirical correlations between salinity and (relative) conductivity that set PSS-78. The standard deviations of

these fits are 0.0007 for atmospheric pressure and 0.0015 for high pressures and correspond to uncertainties of 0.0014 and 0.003, respectively (Le Menn, 2011).

Both the salinities of the samples used to develop the conductivity–salinity relation PSS-78 and the salinities of the samples used to develop the density–salinity relation were thus determined by weighing measurements. If a relation between density and conductivity is set using both relations, then both (relation) uncertainties have be taken into account. It should be noted,

that the density–conductivity relation is only valid, if standard seawater is consistent in its composition. Conversely, this relation can therefore be used to check the standard seawater composition.

The uncertainty in a salinity determined by means of conductivity measurement that is supported by PSS-78 is (in a best-case scenario) 0.0022 using a laboratory salinometer and 0.0034 using a conductivity-temperature-depth device (Le Menn, 2011). These uncertainties are 2 g m$^{-3}$ and 3 g m$^{-3}$ in terms of density. The accuracy of the seawater densities for atmospheric pressure

fulfils these criteria, both in absolute terms and relative to the water reference. In the high-pressure range, it is currently not possible to achieve a comparable accuracy in absolute density using the substitution method and a water reference, as here, the uncertainty in the water reference density is too high. This can be circumvented by stating the seawater density relative to water.

Since the aim of developing the density–salinity relation was to determine the salinity by measuring density with greater

accuracy than by measuring conductivity, a *relative density–salinity relation* was developed instead of an *absolute density–salinity relation*. The accuracy of a salinity that is determined by measuring density at high pressure and subsequent calculation using the (relative) density–salinity relation is thus comparable to the salinity accuracy of conductivity-temperature-depth devices.

### 3 Density corrections

Standard seawater is prepared using natural seawater taken from the North Atlantic Ocean. In order to adjust the required salinity, the natural seawater is diluted with water prepared using groundwater taken from the British mainland; since the groundwater is isotopically depleted, the isotopic water composition of the natural seawater changes during dilution. After preparation, the seawater is poured into borosilicate glass vessels for delivery; these vessels are not completely inert against seawater. Since the seawater was stored in these vessels until the density measurements were made, glass material was

dissolved into the seawater, changing the chemical composition by mainly increasing the dissolved silicate.

For the substitution measurements, the seawater was taken directly from these vessels and pumped into the substitution densimeter, where the temperature is altered; since the seawater was air-saturated at 20 °C in the vessels before being pumped into the densimeter, the air saturation changed in measurements at other temperatures. Since the seawater density is significantly affected by these changes compared to the measurement uncertainty of 2 g m$^{-3}$, it is necessary to apply corrections

to uniform isotopic water and chemical composition, as well as to uniform air saturation.

In this section, corrections for these density effects to the following uniform conditions are presented: the hydrogen–deuterium (H–D) and oxygen-16, 17, and 18 ($^{16}$O–$^{17}$O–$^{18}$O) isotopic composition of VSMOW, the initial chemical composition of the seawater before pouring (especially the silicate content), and air saturation, which depends on temperature. The corrections





presented had been applied to the measured substitution seawater densities before the density–salinity relation was developed, thereby enabling uniform conditions and thus consistency.

### 3.1 Isotopic composition

Water shows a variation in its isotopic composition. The natural variation comprises the H–D relation and the $^{16}$O–$^{17}$O–$^{18}$O

relation. The isotopic abundance of a water sample is usually stated relative to that of the reference material VSMOW, whose isotopic composition is based on a mixture of ocean waters and melted ice (IAEA, 2009). The D isotopic abundance (as well as the $^{18}$O abundance) is usually expressed as the ratio of the amount-of-substance ratio of D and H in the sample to the respective ratio in VSMOW, $\delta_D$:

$$\delta_D = \left[\tfrac{D}{H}\right]^{Sample} / \left[\tfrac{D}{H}\right]^{VSMOW} - 1. \tag{2}$$

The $^{17}$O abundance is usually not monitored, as it is very small compared to the $^{18}$O abundance. In Earth's deep ocean layers, the isotopic composition varies by up to 4 ‰ in D and 0.3 ‰ in $^{18}$O, whereas in the surface ocean layers, these variations are up to 35 ‰ and 3 ‰ (Ferronsky and Polyakov, 2012) due to precipitation. A variation in the isotopic abundance affects the density directly: The corresponding variations are 0.1 g m$^{-3}$ for the deep ocean and 1.3 g m$^{-3}$ for the surface ocean if calculated using Eq. (A.2) given in Appendix A. Isotopic composition variations in the water of the pedosphere are even more significant.

The D and $^{18}$O isotopic abundances $\delta_D$ and $\delta_{18}$ in the natural seawater that was used as the raw material for the diluted seawater preparation at the area of sampling were measured in 1972 and made available by Ostlund et al. (1987). The water, which is purified and used for dilution of the natural seawater, is tap water from Havant, UK, where the supplier of the IAPSO SSW is located. Darling et al. (2003) analyzed the isotopic composition of fresh waters in the British Isles. They used isotope measurement data collected from around 1978 to 2003, including in the region from that the water for dilution was taken. The

relevant values and uncertainties given by Ostlund et al. and Darling et al. are given in Table 2. The equations used to calculate the isotopic abundances of the diluted seawater after mixing standard seawater with water can be derived from the amount-of-substance balance of the isotope considered. For D and $^{18}$O, the equations derived are:

$$\delta_D^{DSW} = \frac{\left(\delta_D^{H_2O}+1\right)\cdot m^{H_2O}+\left(\delta_D^{SSW}+1\right)\cdot m^{SSW}\cdot(1-S_A)}{m^{H_2O}+m^{SSW}\cdot(1-S_A)} \text{ and} \tag{3}$$

$$\delta_{18}^{DSW} = \frac{\left(\delta_{18}^{H_2O}+1\right)\cdot m^{H_2O}+\left(\delta_{18}^{SSW}+1\right)\cdot m^{SSW}\cdot(1-S_A)}{m^{H_2O}+m^{SSW}\cdot(1-S_A)}, \tag{4}$$

where 'DSW' refers to diluted seawater (after mixing) and $S_A = S_R = 35.16504/35 \cdot S_P \cdot$ (g kg$^{-1}$) is the absolute salinity of standard seawater, which is assumed to be equal to the reference salinity of IAPSO SSW according to the recommendation of Millero et al. (2008). Calculated isotopic abundance values and corresponding standard uncertainties of the seawater samples used for the density measurements are given in Table 2. For calculation of the uncertainty, only the isotopic abundances of the water and seawater were taken into account, as the other contributions are insignificant (for example, the salinity of the natural

seawater, which is diluted, may differ by multiple g kg$^{-1}$ without affecting $\delta_D^{DSW}$ and $\delta_{18}^{DSW}$ significantly).

The density difference due to the isotopic abundance change during preparation, $\Delta\rho_{prep}^{SW}$, is estimated using Eq. (A.2), where $\Delta\delta_D = \delta_D^{DSW} - \delta_D^{SSW}$ and $\Delta\delta_{18} = \delta_{18}^{DSW} - \delta_{18}^{SSW}$ are inserted for this purpose. Following this procedure, the isotopic abundance effect on density is assumed to be the same for seawater as for water at $T_{\rho_{max}} = 3.98$ °C and $p_0 = 101325$ Pa and is calculated relative to the isotopic composition of IAPSO SSW. $\Delta\rho_{prep}^{SW}$ is approximated by:

$$\frac{\Delta\rho_{prep}^{SW}(S,T_{\rho_{max}},p_0)}{\text{g m}^{-3}} = -0.0700 \cdot S + 2.4577, \tag{5}$$



where $\Delta\rho_{\mathrm{prep}}^{\mathrm{SW}}(S, T, p) \approx \Delta\rho_{\mathrm{prep}}^{\mathrm{SW}}(S, T_{\rho_{\max}}, p_0)$, and $S$, $T$, and $p$ are the salinity, temperature and (absolute) pressure, respectively. The uncertainty in $\Delta\rho_{\mathrm{prep}}^{\mathrm{SW}}$ is estimated to be 0.3 g m$^{-3}$; uncertainties in the isotopic abundances are insignificant.

$\Delta\rho_{\mathrm{prep}}^{\mathrm{SW}}$ is illustrated in Fig. 2. The more water is used for dilution, the more the density decreases, as the water is depleted in heavy isotopes compared to seawater. The density difference caused by the difference between the isotopic composition of

VSMOW and that of IAPSO SSW (which is given in Table 2), $\Delta\rho_{\mathrm{iso}}^{\mathrm{SW}}$, is 0.3 g m$^{-3}$.

### 3.2 Chemical composition

The seawater used for the measurements was stored in 230 mL borosilicate glass vessels (Bacon, 2007) from the time of preparation at OSIL to the time of measurement. During this time, glass material dissolved into the seawater has significantly altered the chemical composition, and thus the density.

### 3.2.1 Silicate content of standard seawater

Uchida et al. (2011) analyzed the silicate increase in standard seawater delivered by OSIL that was stored in the vessels mentioned above. The silicate increase is related to the dissolution of silica from the glass vessel material. Uchida et al. measured the silicate molality of samples from batches P144 to P152 depending on their storage time. This data was used to estimate the initial silicate molality of the standard seawater used for the density measurements $b_0(S = 35)$ after it had been

prepared, and directly before it was poured into the vessels: 16.5 μmol kg$^{-1}$ with a corresponding estimated uncertainty of 20 %. Note that this silicate molality – which, in terms of conductivity, is insignificant – agrees well with that of standard seawater of batches up to P71 (Poisson et al., 1978) that were analyzed shortly before the conductivity measurements on batch P75 and P79 seawater to develop the PSS-78 equations.

### 3.2.2 Silicate content of the samples used for density measurements

The silicate concentration of some DSW samples from the batches with salinities of 5, 10, 15, 20, 25, and 30 was measured shortly after all density measurements had been performed. The silicate concentration was measured at the *Alfred-Wegener-Institut für Polar- und Meeresforschung* in Bremerhaven, Germany, using an Evolution III flow-through spectrophotometer (Alliance Instruments GmbH, Salzburg, Austria) according to Grasshoff et al. (1999). The device was calibrated before, between and after the DSW sample measurements by measuring Merck Millipore Certipur silicon standard solutions (Merck

KGaA, Darmstadt, Germany), which had a salinity of 36 and reference concentrations of around 7 μmol L$^{-1}$ and 50 μmol L$^{-1}$. The silicate concentration values of the DSW samples were converted to molality values and are given in Table 3, including the corresponding storage time. The silicate molality of the seawater that had salinities of 10, 30, and 35 is higher than that of the other batches, due to the fact that it was stored longer in the vessels (see Table 1 for details).

The reproducibility of a silicate concentration measurement that uses the standards and method described above is usually

within 3 % (K.-U. Ludwichowski, personal communication, 2015). Since the dissolution of the vessel material partly depends on the individual vessel, the difference in the silicate molalities of two measurements (e.g. a salinity of 10) may be higher.

According to Grasshoff et al., the accuracy of the measured silicate concentrations also depends on the difference in salinity between the Certipur standard solutions and the DSW samples. Grasshoff et al. recommend correcting for this effect by applying a constant, device-dependent correction factor derived from calibration measurements. The resulting correction

increases linearly based on the salinity difference between the sample (higher salinity) and the standard (lower salinity). For measurements of samples with a salinity of greater than 30, the correction is smaller than 3 % (AWI, 2015). Assuming a correction due to the salinity effect of 3 % at a salinity difference of 6 and a linear increase thereof, the correction consequently





increases to 10 % at a salinity of 15 and to 16 % at a salinity of 5. We considered this by including the effect in the uncertainty. To this end, we estimated the uncertainty in silicate molalities to be dominated by the batch homogeneity for salinities above 20; for salinities lower than 20 we estimated the uncertainty to be dominated by the correction due to the salinity. Values of the estimated uncertainty in silicate molality are given in Table 3.

### 3.2.3 Density correction to initial silicate content

Since the density measurements obtained from seawater samples were performed before the silicate molality measurements, the storage time (and consequently the silicate molality) were different at that time.

Uchida et al. (2011) estimated the relation between the silicate molality $b$ and the storage time $t$ in the vessels to be linear. The silicate–storage time relation of the seawater samples used in the density measurements is therefore estimated based on the initial silicate molality $b_0$ (of Uchida et al.) and the measurements of the silicate molality $b_1$ (given in Table 3) at storage time $t_1$ given by:

$$b = b_0 + \frac{b_1 - b_0}{t_1}. \tag{6}$$

The initial silicate molality of the DSW samples that have a salinity of less than 35 is derived from $b_0 = S_P \cdot b_0(S_P = 35)/35$, where the water added to the SSW is assumed to be free of silicate.

The borosilicate vessels used to store the of seawater samples are assumed to consist of $w_{SiO_2} = 80$ % (in weight) silica similar to Duran (DURAN Group GmbH, 2009) or Pyrex (Corning Inc., 2014) borosilicate glass. The dissolution of the silica material is determined using the measurements described above. The dissolution of the remaining 20 % borosilicate glass material, which is $B_2O_3$ (13 %) but also $Na_2O$, and $Al_2O_3$, is assumed to be similar to the dissolution of silica (Grambow, 1985). The overall dissolved mass of glass material is consequently given by $\Delta m = M_{SiO_2}/w_{SiO_2} \cdot \Delta n_{SiO_2}$, where $M_{SiO_2} = 60.08$ kg kmol$^{-1}$ is the molar mass of silica and $\Delta n_{SiO_2}$ is the amount-of-substance silica from the glass material that dissolved into seawater (relative to the initial silicate molality). Additionally, $\Delta n_{SiO_2} \approx (b - b_0) \cdot m$, where $m$ is the seawater mass.

The increase in seawater density due to the dissolution of glass material during storage, $\Delta\rho_{stor}^{SW}$, is calculated assuming that the seawater volume remains constant and only the mass increases:

$$\Delta\rho_{stor}^{SW} = \rho \cdot \frac{M_{SiO_2}}{w_{SiO_2}} \cdot (b - b_0), \tag{7}$$

where $\rho$ is the seawater density. The uncertainty in the density correction due to the dissolution of glass material is estimated using Eq. (7) as a model equation, with Eq. (6) being inserted. Furthermore, the following uncertainties are considered: (i) uncertainty in the silica mass fraction of glass material: 5 %, (ii) uncertainty in the initial silicate content $b_0$: 20 %, (iii) uncertainty in the measured silicate content $b_1$: as given in Table 3, and (iv) uncertainty in the storage time $t_1$: 15 days.

Some values of the density correction that were applied to the measured seawater densities are shown in Fig. 3. The corrections are about 1 g m$^{-3}$ to 3 g m$^{-3}$ and the corresponding estimated uncertainties are 0.4 g m$^{-3}$, which yields an increase in uncertainty of the measured values at atmospheric pressure of up to 8 %. The scatter of the correction values for high pressures is higher than that for atmospheric pressure, as density measurements at high pressures take significantly longer; as a result, the period between the first and last measurement is longer as well.





### 3.3 Air saturation

Usually, seawater samples used in highly accurate density measurements in laboratories are air-saturated, as any degassing procedure may change the salt composition. For water, the effect of air solubility on density has been measured directly, e.g. by Bignell (1983) through comparing the densities of saturated and desaturated water.

For our density measurements, the seawater samples were taken directly from the vessels delivered by OSIL as shown in Figure 1a. The vessels were stored in our laboratory at a temperature of approximately 20 °C, at which the seawater equilibrated with the air inside the (closed) vessels. Since the seawater was also pumped into the VTD at this temperature, the air saturation was 100 % at 20 °C. After filling the VTD, the seawater temperature was altered to the measurement temperature. During this time, the saturation changed to undersaturation at temperatures lower than 20 °C and to oversaturation at

temperatures higher than 20 °C, as there was no contact to air during the time of temperature equilibration, which is approximately 15 min. This temperature-dependent gassing is significant compared to the density measurement uncertainty. For consistency of the air saturation, the measured densities have to be corrected to a saturation of either 0 % or 100 %. Because the density corrections to 100 % are significantly smaller than those to 0 %, and because any degassing procedure is problematic, the density values were corrected to 100 % air saturation. Following this procedure, the density–salinity relation

was developed with the least loss in accuracy.

The density correction is estimated taking into account the fact that the amount of air molecules remains constant while the liquid temperature changes from 20 °C to measurement temperature before density measurement. A complex calculation similar to that given by Harvey et al. (2005) for the air saturation of water was used to quantify the density change of seawater due to saturation. For this calculation, the partial molar volumes of nitrogen, oxygen, argon, and carbon dioxide in water were

assumed to be equal in seawater. Salinity-dependent solubility data of nitrogen and argon were taken from Hamme and Emmerson (2004), of oxygen from Garcia and Gordon (1992), and of carbon dioxide from Weiss (1974). The calculation showed that the different gas solubilities in water and seawater are negligible in terms of density, as the deviation between the calculated density change of seawater and that of water (of Harvey et al.) is around 0.1 g m$^{-3}$. Furthermore, it was found that it is sufficient to consider only the nitrogen solubility to calculate the density correction that is approximated by:

$$\Delta\rho_{\text{aer}}^{\text{SW}} = \left(1 - \frac{n_{\text{N}_2}(100\,\%,\,20\,°C)}{n_{\text{N}_2}(100\,\%,\,T)}\right) \cdot \Delta\rho_{\text{a}}^{\text{H}_2\text{O}}(100\,\%,\,T), \tag{8}$$

where $n_{\text{N}_2}(100\,\%,\,20\,°C)$ and $n_{\text{N}_2}(100\,\%,\,T)$ are the dissolved nitrogen amounts of substance at 100 % saturation at 20 °C and at measurement temperature as well as $\Delta\rho_{\text{a}}^{\text{H}_2\text{O}}(100\,\%,\,T)$ being the corresponding density effect whose calculation is described in Appendix A.

At measurement temperatures higher than 20 °C, the seawater is oversaturated during density measurement, as it was saturated

at 20 °C before filling. It is assumed that the nucleation of microbubbles due to the oversaturation takes significantly longer than the time of temperature stabilization and density measurement, which is always less than 30 min. The density effect caused by oversaturation is therefore assumed to be proportionally equal to that up to saturation. The calculated density correction and the corresponding estimated uncertainty, which is 0.4 g m$^{-3}$, are illustrated in Fig. 4. The density correction is significant at temperatures less than 15 °C compared to the measurement uncertainty of 2 g m$^{-3}$, as the gas solubility is

significantly higher at low temperatures.

Based on a measured substitution density, $\rho_{\text{subs}}^{\text{SW}}$, that has been corrected to the uniform isotopic water and chemical compositions and to 100 % air saturation, a seawater density, $\rho^{\text{SW}}$, was calculated by means of:

$$\rho^{\text{SW}} = \rho_{\text{subs}}^{\text{SW}} - \Delta\rho_{\text{prep}}^{\text{SW}} + \Delta\rho_{\text{iso}}^{\text{SW}} - \Delta\rho_{\text{stor}}^{\text{SW}} + \Delta\rho_{\text{aer}}^{\text{SW}} + \Delta\rho_{\text{tar}}^{\text{SW}}, \tag{9}$$

where $\Delta\rho_{\text{tar}}^{\text{SW}}$ is a density correction to integer salinities introduced for practicability.



## 4 Density–salinity relation

In this section, the development of the density–salinity relation is described. Although this relation should be used to determine the salinity by means of density, it was set up as a density function of salinity, temperature, and (absolute) pressure, i.e. $\rho = f(S, T, p)$, as doing so allows the data to be approximated more precisely. As a result, the salinity has to be calculated using

inverse methods. Since the relation was developed relative to the water density for higher accuracy, the salinity range from 0 to 5 is included by adding the values of pure water. In order to make use of this relation even beyond this range and the ranges measured, the uncertainty was estimated for somewhat wider ranges in the absence of measurement data. The relation accuracy was verified by means of a new method that verifies the uncertainty in predicted results locally using the measurement results, taking into account the correlation between the two. This is particularly advantageous for empirical fit equations, as these are

not physical laws and are therefore not inherently consistent, i.e. they are not independent of the measurement results themselves.

### 4.1 Physical model

The density of air-saturated seawater is modelled based on degassed water, whose density is given by $\rho_0^{\mathrm{H_2O}}$. Salt that has a relative composition similar to that dissolved in standard seawater is added to the degassed water. The salt content is given

implicitly by the salinity. The density of the degassed water changes after the salt is added by $\Delta\rho_0^{\mathrm{SW}}$. In addition, air with a defined composition is absorbed, as a result of which the density changes by $\Delta\rho_a^{\mathrm{SW}}$. The density of air-saturated seawater, $\rho^{\mathrm{SW}}$, is thus given by:

$$\rho^{\mathrm{SW}} = \rho_0^{\mathrm{H_2O}} + \Delta\rho_0^{\mathrm{SW}} + \Delta\rho_a^{\mathrm{SW}}, \tag{10}$$

where $\rho_0^{\mathrm{H_2O}}$ is the density of degassed water, $\Delta\rho_0^{\mathrm{SW}}$ is the density change due to dissolved salt, and $\Delta\rho_a^{\mathrm{SW}}$ is the density change

due to absorbed air. The density change due to dissolved salt and absorbed air may be summarized by $\Delta\rho^{\mathrm{SW}}$ and may also be called relative density of air-saturated seawater, as the seawater density was measured relative to water in the substitution measurements.

If the salt is added at the atmospheric pressure $p_0$, the water density changes by $\Delta\rho_0^{\mathrm{SW}}(p_0)$. If the salt is added at the pressure $p \neq p_0$, the water density changes by $\Delta\rho_0^{\mathrm{SW}}(p)$. If the difference between the two changes is $\Delta\Delta\rho_0^{\mathrm{SW}}(p - p_0)$, then the density

change due to dissolved salt at any pressure is given by:

$$\Delta\rho_0^{\mathrm{SW}} = \Delta\rho_0^{\mathrm{SW}}(p_0) + \Delta\Delta\rho_0^{\mathrm{SW}}(p - p_0), \tag{11}$$

where $\Delta\rho_0^{\mathrm{SW}}(p_0)$ is the density change due to dissolved salt at the atmospheric pressure $p_0$ and $\Delta\Delta\rho_0^{\mathrm{SW}}(p - p_0)$ is the difference between the density changes at the pressure $p$ and at the atmospheric pressure $p_0$.

The solubility of gases in liquids is well described at infinite dilution and low pressure by means of the Henry law, according

to which the number of absorbed gas molecules is proportional to the gas pressure above the liquid. However, since there is no reservoir for additional gas at high pressure, only the air absorption at the gas pressure $p_0$ is taken into account for modelling. In addition, it is assumed that the absorbed air is incompressible. In the model, the density change due to absorbed air is therefore not treated as a function of pressure, i.e. $\Delta\rho_a^{\mathrm{SW}} \neq f(p)$.

Air solubility in seawater depends on salinity (Hamme and Emmerson, 2004; Garcia and Gordon, 1992). The solubility in

water and seawater was compared above. Based on this, it can be assumed that the resulting density change of water and seawater is approximately equal. In the model, the density change due to absorbed air is therefore not a function of the salinity either, i.e. $\Delta\rho_a^{\mathrm{SW}} = \Delta\rho_a^{\mathrm{H_2O}} \neq f(S)$.




### 4.2 Fitting of $\Delta\rho_0^{SW}(p_0)$

The values of the seawater density for atmospheric pressure, $\rho^{SW}$, which were obtained from the measurements and corrected to the uniform conditions, were broken down according to Eq. (10) into the corresponding values of the water density, $\rho_0^{H_2O}$, and the values yielded by the density change due to dissolved salt and absorbed air (or relative density of air-saturated

seawater), $\Delta\rho^{SW}$. For this purpose, the water density was calculated using the equation of state developed by Wagner and Pruß (2002), by means of which the water reference density for the substitution measurements was calculated as well. Therefore, the uncertainty in the relative density is up to 20 % lower than that in the absolute density.

The values of the relative density of air-saturated seawater $\Delta\rho^{SW}$ were broken down into the resulting values of the density change due to dissolved salt, $\Delta\rho_0^{SW}(p_0)$, and the corresponding values of density change due to absorbed air, $\Delta\rho_a^{SW}$. For this

purpose, the values of $\Delta\rho_a^{SW}$ were calculated using the equation of Harvey et al. (2005), which is valid for the absorption of air into water at $p_0 = 101325$ Pa, but were adopted for the absorption of air into seawater according to the physical model described above:

$$\frac{\Delta\rho_a^{SW}(T)}{g\,m^{-3}} = 0.103 - 2.371 \times 10^5 \cdot \left(\frac{T}{°C} + 75\right)^{-2.5} + 1.82 \times 10^{-7} \cdot \left(\frac{T}{°C} + 75\right)^3, \qquad (12)$$

where the model air composition is 78.1 % $N_2$, 20.9 % $O_2$, 0.9 % Ar, and 0.4‰ $CO_2$. Note that this equation is also given in

Appendix A, but is repeated here for clarity.

The values of the relative density of degassed seawater, $\Delta\rho_0^{SW}(p_0)$, were used to fit the coefficients $a_{i,j}$ of the following empirical equation:

$$\Delta\rho_0(p_0) = \Delta\rho_0^o \cdot \sigma \cdot \sum_{i=0}^{5}\sum_{j=0}^{5-i} a_{i,j} \cdot \tau^i \cdot \sigma^j, \qquad (13)$$

where $\Delta\rho_0^o = 30$ kg m$^{-3}$, $\tau = T/T^o$ is the reduced temperature with $T$ being the temperature in K and $T^o = 288.15$ K, $\sigma =$

$S/S^o$ is the reduced salinity with $S$ being the salinity and $S^o = 35$. Note, that the values of $\Delta\rho_0^o$, $T^o$, $S^o$ (as well as $\Delta\Delta\rho_0^o$ and $\pi^o$ below) were chosen for practical handling of the fit coefficient values and do not have a physical meaning.

The linear fit coefficients $a_{i,j}$ were determined by uncertainty-weighted least squares fitting within the Monte Carlo based approach described in Appendix B. The fit coefficients were initially averaged from up to $n = 1,500$ runs, where no longer significant effects on calculated values or uncertainties thereof were found. Finally, the coefficients were averaged from $n =$

15,000 runs for safety reasons. The fitting yielded the values of $a_{i,j}$ given in Table 4, which were reduced to the significant number of digits.

The residuals of the fit using the coefficients given in Table 4 are illustrated in Fig. 5, where they are compared with the density–salinity relation uncertainty, $U$, whose determination is described below. The fit standard deviation is 1.1 g m$^{-3}$. No systematic deviation of the residuals depending on salinity or temperature was found.

If the density of air-saturated seawater is calculated using the density–salinity relation, the (fitted) relative seawater density plus the (artificially inserted) density change due to absorbed air is used, i.e. $\Delta\rho^{SW} = \Delta\rho_0^{SW} + \Delta\rho_a^{SW}$. However, $\Delta\rho_a^{SW}$ has practically no statistical influence on the fitting of $\Delta\rho_0^{SW}$, and therefore no statistical influence on $\Delta\rho^{SW}$. Consequently, if $\Delta\rho^{SW}$ is calculated, its uncertainty is $U(\Delta\rho_0^{SW})$, i.e. that of the degassed seawater density, whereas, if $\Delta\rho^{SW}$ is calculated, its uncertainty is $\left[U(\Delta\rho_0^{SW})^2 + U(\Delta\rho_a^{SW})^2\right]^{1/2}$, i.e. that of the degassed seawater density and that of the density change due to

absorbed air.

The uncertainty in $\Delta\rho_0^{SW}(p_0)$ was determined and verified using the approach described in Appendix B. The calculated uncertainty is at least (0.7 g m$^{-3}$) at a salinity of 15 and at 25 °C, and increases as expected at higher salinities, as well as at





lower and higher temperatures (up to 1.2 g m$^{-3}$). The subsequent uncertainty verification yielded four inconsistent densities whose residuals were higher than their corresponding uncertainties. The uncertainty was therefore increased to 2 g m$^{-3}$ in the entire measurement region of salinities up to 35 and temperatures from 5 °C to 35 °C.

Since the density–salinity relation may be used for calculations in a wider region, e.g. salinities up to 40 and temperatures from
0 °C to 40 °C, we also estimated the uncertainty in $\Delta\rho_0^{SW}(p_0)$ for this region in the absence of measurement data. The density uncertainty in the wider (extrapolation) region was also calculated using the approach described in Appendix B, whereby the possible variation of the fit polynomial outside the measured salinity and temperature region is taken into account. The uncertainties resulting from this calculation are shown in Fig. 6a together with the uncertainty of the measurement region. For practicability, the highest uncertainty in a particular region was assigned. The uncertainties in the extrapolation region are at
least twice as much as in the measurement region.

For calculating salinity using relative density and temperature values by means of the density–salinity relation, the uncertainty in salinity was also determined in the measurement and extrapolation region. The salinity uncertainty was calculated by multiplying the density uncertainty by the partial derivative of salinity by density, i.e. $U(S) = U(\Delta\rho^{SW}) \cdot \partial S/\partial \rho$. The uncertainties resulting from this calculation are shown in Fig. 6b. A salinity determined by means of a calculation using the
relation has an uncertainty of $3 \times 10^{-3}$. Note that, if measurement values are used for calculation, their contribution also has to be included.

Since no uncertainty verification in the extrapolation region is possible using the fitting data set, additional substitution density measurements obtained from some samples of the seawater used for determination of the density–salinity relation were conducted. The results of these measurements, which were performed at 1 °C and at atmospheric pressure, were corrected to
the uniform isotopic water and the chemical salt compositions as well as air saturation as described in Sect. 3. The density deviations of these corrected results from the predicted values of the density–salinity relation are shown in Fig. 7. The deviations are well within the uncertainty of 4 g m$^{-3}$ assigned to this region.

### 4.3 Fitting of $\Delta\Delta\rho_0^{SW}(p - p_0)$

The values of the seawater density for high pressures, $\rho^{SW}$, were broken down according to Eq. (10) into corresponding values
of the water density, $\rho_0^{H_2O}$, and the values yielded by the density change due to dissolved salt and absorbed air (or relative density of air-saturated seawater), $\Delta\rho^{SW}$. Since the water density was calculated analogously to the atmospheric pressure densities, the uncertainty in the relative density is up to 50 % lower than that in the absolute density for pressures up to 10 MPa and up to 80 % lower for up to 65 MPa.

The values of the relative density of air-saturated seawater, $\Delta\rho^{SW}$, were broken down into the values yielded by the density
change due to dissolved salt, $\Delta\rho_0^{SW}$, and the corresponding values of the density change due to absorbed air, $\Delta\rho_a^{SW}$. For this purpose, the values of $\Delta\rho_a^{SW}$ were calculated analogously to the atmospheric pressure densities using Eq. (12).

The values of the density change due to dissolved salt, $\Delta\rho_0^{SW}$, were further broken down according to Eq. (11) into the values of the density change due to dissolved salt at the atmospheric pressure $p_0 = 101325$ Pa, $\Delta\rho_0^{SW}(p_0)$, and the difference between the density change at (high) pressure $p$ and that at the pressure $p_0$, $\Delta\Delta\rho_0^{SW}(p - p_0)$. The relative density values for atmospheric
pressure that had been used to fit the coefficients of Eq. (13), were used for this purpose.

The resulting values of the density difference $\Delta\Delta\rho_0^{SW}(p - p_0)$ were used to fit the coefficients $b_{i,j,k}$ of the following empirical equation:

$$\Delta\Delta\rho_0^{SW}(p - p_0) = \Delta\Delta\rho_0^o \cdot \sigma \cdot \pi \cdot \sum_{i=0}^{4}\sum_{j=0}^{4-i}\sum_{k=0}^{4-i-j} b_{i,j,k} \cdot \tau^i \cdot \sigma^j \cdot \pi^k, \tag{14}$$





where $\Delta\Delta\rho_0^{\mathrm{o}} = 2$ kg m$^{-3}$, $\pi = (p/p^{\mathrm{o}} - 1)/\pi^{\mathrm{o}}$ with $p$ being the pressure in MPa, $p^{\mathrm{o}} = p_0 = 0.101325$ MPa and $\pi^{\mathrm{o}} = 1000$. Due to the formulation of the dimensionless pressure $\pi$, $\Delta\Delta\rho_0$ is exactly zero at $p_0$, thereby ensuring the high accuracy of the density at atmospheric pressure $\Delta\rho_0(p_0)$.

The linear fit coefficients $b_{\mathrm{i,j,k}}$ were determined analogously to the fit coefficients $a_{\mathrm{i,j}}$ using the approach described in

Appendix B. The fitting yielded the values of $b_{\mathrm{i,j,k}}$ given in Table 5, which were reduced to the significant number of digits. The residuals of the fit using the coefficients given in Table 5 are illustrated in Fig. 8, where they are compared with the density–salinity relation uncertainty whose determination is described below. The fit standard deviation is 2.3 g m$^{-3}$. At a pressure of 65 MPa two residuals exceeds the uncertainty significantly. No systematic deviation of the residuals depending on salinity, temperature, or pressure was found.

The data set for fitting $\Delta\Delta\rho_0^{\mathrm{SW}}(p - p_0)$ comprises pressures up to 65 MPa. Since the density–salinity relation may be used for calculations over a wider range, e.g. pressures up to 100 MPa, we also estimated the uncertainty in this range in the absence of measurement data.

The density uncertainty in the measurement region was determined using the approach described in Appendix B. Summarized results of this calculation are shown in Fig. 9a and 9c together with the results of the measurement region. For practicability,

the highest uncertainty in a particular region was assigned. The uncertainties in the extrapolation region are at least twice as much as in the interpolation region.

For calculating salinity using relative density, temperature, and pressure values by means of the density–salinity relation, the uncertainty in salinity was also determined in the interpolation and extrapolation region. The salinity uncertainty was calculated by multiplying the density uncertainty by the partial derivative of salinity by density, i.e. $U(S) = U(\Delta\rho^{\mathrm{SW}}) \cdot \partial S/\partial\rho$. The

uncertainties yielded by this calculation are shown in Fig. 9b and 9d. A salinity determined by means of a calculation using the relation has an uncertainty of $8 \times 10^{-3}$. Note that, if measurement values are used for calculation, their contribution also has to be included.

## 5 Comparison with TEOS-10

The present reference equation of state for thermodynamic properties of seawater is the Thermodynamic Equation of Seawater

TEOS-10 adopted by the Intergovernmental Oceanographic Commission (IOC et al., 2010). TEOS-10 describes the properties of degassed seawater in wide ranges of salinity, temperature, and pressure relative to degassed water with the VSMOW isotopic composition. Relative density values calculated using TEOS-10 with salinities from 0 to 40 and temperatures from 0 °C to 40 °C have estimated uncertainties of 8 g m$^{-3}$ for atmospheric pressure, 17 g m$^{-3}$ up to 10 MPa, and 26 g m$^{-3}$ up to 100 MPa. In order to possibly reduce the estimated density uncertainty in these regions, TEOS-10 was compared with the density–salinity

relation.

### 5.1 Atmospheric pressure

For atmospheric pressure, the density deviation of TEOS-10 from the density–salinity relation is shown in Fig. 10a. TEOS-10 density values are always higher than those of the density–salinity relation. The increase of the deviation with salinity is approximately linear. At salinities higher than 25, the deviation exceeds the estimated uncertainty of 8 g m$^{-3}$ significantly. At

salinities smaller than 5, the deviation, although consistent, is unexpectedly high. Salinity 0, which is pure water, defines the zero-line of TEOS-10 and of the density–salinity relation.





In order to leave the linear increase of the deviation with salinity seen in Fig. 10a out of consideration, a reduced form is shown in Fig. 10b. Here, $\Delta\rho - \Delta\rho(S = 35) \cdot S/35$ is visualized. It is found that the reduced deviation is always less than 5 g m$^{-3}$.

In order to find possible causes for the unexpectedly high density deviation, the density data on which TEOS-10 is based, were examined; the uncertainty in salinity was considered negligible. In the $(S, T, p_0)$-region of interest, according to Feistel (2003

and 2008), TEOS-10 is based on a dataset (JPOTS, 1981c, pp. 36–56) that consists of normalized density data of Millero et al. (1976) and of Poisson et al. (1980), where the density data of Millero et al. has a significantly higher precision. Note that, for atmospheric pressure, this dataset was also used to fit the former reference equation of state EOS-80 (JPOTS, 1981c), and, therefore, no comparison with EOS-80 was carried out.

Millero et al. measured the density of diluted and standard seawater of batch P63 using a magnetic float densimeter. The

comparison between the normalized densities measured by Millero et al. and TEOS-10 shown in Fig. 11a suggests that TEOS-10 is well fitted to these densities. A comparison between the densities measured by Millero et al. and the density–salinity relation is shown in Fig. 11b. Here as well, a systematically linear density deviation with salinity is found; the jump between the deviations at salinities greater than or equal to 30 and the other salinities is especially noticeable. In addition, the deviations here are the least scattered. For calculation of the density deviations given in Fig. 11a and b, the temperatures at that Millero

et al. made their measurements were converted from the International Practical Temperature Scale 1968 to the International Temperature Scale 1990 (CCT, 1997).

Since the seawater samples with salinities of less than 35 measured by Millero et al. were prepared by means of weight dilution, it is unlikely that the salinity determination is a cause for the systematically appearing deviation and the high scatter, which is also present in the comparison with TEOS-10 shown in Fig. 11a. It is therefore likely that the accuracy and precision of the

seawater density measurement of 2 g m$^{-3}$ and 1 g m$^{-3}$ given by Millero et al. is overestimated. For this reason, the magnetic flotation method used by Millero et al. was examined in detail for possible issues.

Magnetic float densimeters have an advantage over hydrostatic weighing densimeters: No mechanical coupling by means of a suspension is needed to determine the buoyance force acting on a float (or sinker). Instead, this is achieved with a magnetic coupling by placing a magnet into a float. The float is brought to mechanical equilibrium, i.e. floats in the liquid, by means of

a current-carrying coil; here, the current is a measure of the force, and thus of the liquid density. However, for density measurement the characterisation of the magnetic coupling is necessary in addition to the determination of the float volume, as in case of a hydrostatic weighing densimeter.

The densimeter used by Millero et al. for measuring the seawater density consisted of a hollow float in the measuring liquid of a vessel that had a volume of 250 mL, with the coil mounted underneath. The float was made of Pyrex, contained a permanent

magnet that was a stirring bar and was therefore probably made of Alnico, and had a volume of 32 cm$^{-3}$ (Millero, 1967). The float was weighted with platinum weights to adjust its buoyancy. The current that passed through the coil was used to pull the float to the bottom of the measuring vessel. Subsequently, the current intensity was gradually reduced until the float lifted off the bottom. The equilibrium current determined in this way, which was assumed to define the state of floating, was a measure of the liquid density.

Bignell (2006) discussed various methods for determining the buoyancy force in magnetic float densimeters. For the design of the magnetic coupling system, the magnetic force exerted on a permanent magnet by a current-carrying, circular coil (without a metal core) was given by:

$$F_{\text{mag}} = m \cdot G(z, R) = m \cdot \frac{-3}{2} \cdot \mu \cdot \frac{R^2 \cdot z}{\sqrt{(R^2 + z^2)^5}} \cdot I, \tag{15}$$

where $m$ is the magnetic momentum; $G(z, R)$ is the magnetic field gradient (along the axis perpendicular to the coil plane

through the coil centre point), $\mu$ is the permeability of the medium between the permanent magnet and the coil, $R$ is the circular





coil radius, $z$ is the distance between the magnet and the coil, and $I$ is the current. In a measurement obtained from seawater, the magnetic force is therefore dependent on the magnetic water properties and on the magnet-coil distance.

Bignell pointed out that the force on the magnet is also dependent on the magnetic field, even for magnetically hard materials. The magnetic force is therefore not linearly (as in Eq. 15) but quadratically dependent on the equilibrium current, i.e. $F_{mag} =$

$f_1 \cdot I + f_2 \cdot I^2$, where $f_1$ and $f_2$ are magnetic coupling constants. For a magnetically hard material, the force mainly depends on the linear term, whereas the quadratic term is used as a correction.

Millero et al. used a cylindrical (instead of a circular) coil and summarized the magnetic force as $F_{mag} = f \cdot I$, where the calibration factor $f$ was determined with measurements obtained from air-saturated water by weighing the float with platinum weights. The seawater density was determined relative to water, i.e. relative to the calibration using water:

$$\Delta\rho^{SW} = \frac{f \cdot (I^{SW} - I^{H_2O})}{V + m_{Pt}/\rho_{Pt}},$$   (16)

where $I^{SW}$ and $I^{H_2O}$ are the currents resulting from the measurements obtained from air-saturated seawater and water, $V$ is the float volume, which is also determined by the calibration, $m_{Pt}$ and $\rho_{Pt}$ are mass and density of the platinum weights, which were identical in a measurement obtained from seawater and water.

Since seawater and water have different magnetic properties, it is possible that the calibration factor $f$ is significantly different,

i.e. $\mu^{SW} \not\approx \mu^{H_2O} \Longrightarrow f^{SW} \not\approx f^{H_2O}$. In order to rule out this possibility, a representative calculation was carried out. Since, theoretically, $F_{mag} \propto \mu \cdot I$ for a cylindring (and circular) ring coil, it follows directly that $\mu^{SW}/\mu^{H_2O} = f^{SW}/f^{H_2O}$ if the permanent magnet is in the same position in both measurements; the calibration factor of seawater is thus calculated from that of water. The permeabilities are calculated by $\mu = \mu_0 \cdot (1 + \chi)$, where $\mu_0 = 4 \cdot \pi \cdot 10^{-7}$ N A$^{-2}$ is the vacuum permeability, $\chi^{SW} = -8.25 \times 10^{-6}$ and $\chi^{H_2O} = -9.04 \times 10^{-6}$ are the (dimensionless) volume susceptibilities of seawater with a salinity of 29

(Imhmed, 2012) and of water. The relative density deviation due to the different permeabilities being neglected was calculated by $\Delta\rho^{SW}(f^{SW}, f^{H_2O}) - \Delta\rho^{SW}(f = f^{H_2O})$ using (i) the calibration factor for water $f = f^{H_2O} = -3.5308$ g A$^{-1}$ for 25 °C (Millero, 1967), (ii) the currents $I^{SW} = 0.4$ A and $I^{H_2O} = 0.15$ A, (iii) the platinum mass and density $m_{Pt} = 0.7$ g and $\rho_{Pt} = 21450$ kg m$^{-3}$, and (iv) the float volume given above. Note that (ii) and (iii) were chosen based on a plot of calibration data of the flotation densimeter given by Millero (1967), and correspond to a relative sea water density of 28 kg m$^{-3}$. The calculation

yields a density deviation at the order of 0.01 g m$^{-3}$; as a result, the differences in the magnetic properties of seawater and water are not problematic.

Since the volume of the float was also determined by means of the calibration measurement using water, it is possible that this resulted in a significant deviation in the relative seawater density. A further representative calculation was therefore carried out with the values (i–iv). Using this calculation, a density deviation of only 3 g m$^{-3}$ is yielded for a relative volume deviation

of $10^{-4}$. Although the volume results indirectly from an extrapolation of the linear relation of the magnetic coupling, $F_{mag} = f \cdot I$, which is quadratic even for magnetically hard materials according to Bignell, it is unlikely that a volume deviation of this magnitude will occur in the calibration measurement; the float volume calibration is therefore not problematic.

A final calculation was performed to estimate how significant the precise height positioning of the permanent magnet is, i.e. the distance from the coil. Two reasons for a change of the distance are conceivable. On the one hand, the position of the

magnet (inside the float) or of the coil can change in the time between the calibration measurement obtained from water and the measurement obtained from seawater; the permanent magnet was fixed in the hollow float using wax (Millero, 1967). Density deviations that result from such position changes are minimized if, after each measurement obtained from seawater, a measurement obtained from water had also been carried out (a quasi-substitution measurement). On the other hand, the "lift-off" process, wherein the equilibrium current is determined by sight, is not the same for seawater and water in terms of speed



(among other factors). Density deviations that result from such dissimilarities are minimized, if, in additional to the "lift-off" current, the "drop-down" current had been determined in the opposite manner and both currents had been averaged for seawater and water, respectively. Or, if in the measurement obtained from seawater, the float was weighted with the aim to yield the same current as in the calibration measurement using water.

For the calculation, it was assumed that the height dependence of the magnetic force given by $z$ in Eq. (15) for the circular coil is similar for the cylindrical coil used by Millero et al. If the distance between magnet and coil is $z + \Delta z$, then $f(z + \Delta z)/f(z) = (z + \Delta z)/z \cdot [(R^2 + z^2)/(R^2 + (z + \Delta z)^2)]^{5/2}$ holds. The displacement of the coil or of the magnet can be treated mathematically as the same, since $f^{SW} = f(z + \Delta z)$ applies to the measurement obtained from seawater and $f^{H_2O} = f(z)$ applies to the measurement obtained from water in both cases. Using the values (i–iv), the coil radius $R = 20$ mm and

the distance $z = 40$ mm for an unconsidered distance increase of $\Delta z = 3$ μm yields a relative seawater density which is too high by 10 g m$^{-3}$. Note that $R$ and $z$ were estimated based on a sketch and a dimension of the flotation densimeter used (Millero, 1967). If a temporal or permanent distance increase exists that is not considered, an approximately linear density increase (or decrease) as seen in Fig. 11b results.

  The high sensitivity of the measurement density to the magnet height position is one reason why magnetic flotation densimeters

that were developed later and that share a similar principle, e.g. that of Bignell (1982), use position sensing systems with accuracies that are at least in the micrometer range, in order to keep the height, $z$, constant. The actual cause of the significance of the density deviations seen in Figs. 12a and b may therefore be an overestimation of the accuracy and precision of the magnetic flotation method used.

### 5.2 High pressure

TEOS-10 may be used to calculate densities for pressures up to 100 MPa. In the $(S, T, p > p_0)$-region of interest, the density data given by Chen and Millero (1976), as well as thermal expansion data given by Bradshaw and Schleicher (1970) and speed-of-sound data given by Del Grosso (1974) were used for fitting (Feistel, 2003 and 2008). Chen and Millero directly measured the seawater density, i.e. the specific volume, relative to water using a magnetic float densimeter whose magnetic force on the float was determined as described above. By contrast, the data of Bradshaw and Schleicher, and of Del Grosso allows only the

calculation of difference densities (relative to a reference state with defined salinity, temperature, and pressure) using thermodynamic relations.

  An overview of the density deviation of TEOS-10 from the density–salinity relation in the entire salinity-temperature region for atmospheric pressure is given in Fig. 12a. The increase in the deviation with salinity seen in Fig. 11 for 5 °C, 20 °C, and 35 °C is also present at 0 °C. For higher temperatures and salinities of around 20, the deviation increases unexpectedly. A

similar overview of the density deviation for 30 MPa is given in Fig. 12b. The density deviation of this pressure is higher than that of atmospheric pressure. In the measurement region, this trend continues globally for up to 65 MPa as seen in Fig. 12c, but, in the extrapolation region, discontinues locally for up to 100 MPa as seen in Fig. 12d. For all pressures, the densities calculated using TEOS-10 are higher than the densities calculated using the density–salinity relation. The uncertainty in the deviations, however, is not exceeded significantly for higher pressures, in contrast to atmospheric pressure. For this reason,

further comparisons with the measurement data were not performed.





## 6 Summary

A density–salinity relation for IAPSO standard seawater was developed by means of highly accurate density measurements performed using a recently developed substitution method. This relation makes it possible to consistently determine (practical) salinity by means of density measurement at a level of accuracy that is comparable to that achieved by means of a conductivity

measurement supported by PSS-78 and related application routines. The relation has been developed as a function of salinity, i.e. $\Delta\rho = f(S, T, p)$, relative to the density of water, as such a function was better fitted to the measurements, thereby increasing the accuracy of the predicted results. The relation is valid for seawater with the chemical salt composition of IAPSO standard seawater, for the isotopic water composition of Vienna Standard Mean Ocean Water, and for an air saturation of 100 % at all temperatures and at atmospheric pressure. The reference density is that of degassed water. The measurement range comprises

$0 \leq S \leq 35$, $5\,°C \leq T \leq 35\,°C$, and $0.1\,MPa \leq p \leq 65\,MPa$. In this range, the uncertainty in salinity (calculated from density) is 0.003 for atmospheric pressure and 0.008 for high pressures; the uncertainty in density (calculated from salinity) is $2\,g\,m^{-3}$ and $6\,g\,m^{-3}$, respectively. Since the conditions occurring in the ocean cover a wider range, the relation range of validity has been extended to $0 \leq S \leq 40$, $0\,°C \leq T \leq 40\,°C$, and $0.1\,MPa \leq p \leq 100\,MPa$. In this range, the uncertainty was estimated to be a multiple of that in the measurement range, i.e. usually twice as much. A validation for temperatures down to

$0\,°C$ was performed using additional density measurements.

Density corrections for standard seawater were developed. Because the chemical composition was changed by interactions with borosilicate glass material of the storage vessel, and because the seawater samples used in the measurements were stored for different periods, the measured densities were corrected to a uniform (i.e. the original) chemical composition. These corrections are up to $3\,g\,m^{-3}$. Because the isotopic water composition of the standard seawater changed due to the addition of

water (with less deuterium, oxygen-17 and 18) in the preparation of dilute seawater samples, the measured densities were corrected to the uniform isotopic composition of VSMOW. These corrections are up to $2.5\,g\,m^{-3}$. A further density correction was developed to correct the seawater air saturation to 100 %; where the temperature changed while air was excluded, the corrections were up to $1.5\,g\,m^{-3}$. Taken together, all corrections total more than $5\,g\,m^{-3}$.

The density–salinity relation was compared with the reference equation of state for seawater TEOS-10. For atmospheric

pressure, density deviations of up to $15\,g\,m^{-3}$ were found, which is significantly greater than the deviation uncertainty. Moreover, a systematic, linear dependence on salinity was found. One reason for the deviations is likely an overestimation of the accuracy of the density data that TEOS-10 is based on in this region. For high pressures, density deviations of up to $40\,g\,m^{-3}$ were found, which is of the same order of magnitude as the deviation uncertainty.

## 7 Conclusions

Seawater is changed during storage. Mainly silicon dioxide dissolves from borosilicate glass material and forms silicic acid, but over the long term, the solubility of other glass components is also important. This affects the density of stored seawater. If standard seawater is to be used as a density reference material, the solubility of all glass components must be quantified so that the change in the chemical composition and in density can be calculated. This also includes the dependence of this solution on temperature during storage; storage at low temperatures may minimize this interaction. For long-term storage, container

materials that have a greater chemical resistance should be investigated.

Knowledge of the isotopic composition is essential for measurements obtained from seawater samples that are artificially diluted with water from different locations, as the local isotopic water composition varies significantly. For natural seawater, this may be important in marginal seas.



The data situation of recent highly accurate density measurements on standard seawater is poor, which is why further measurements should be carried out using state-of-the-art methods. The data of the density–salinity relation obtained in the present study should be used as a correction to TEOS-10.

Salinity is usually measured by means of a salinometer measuring conductivity and being calibrated by standard seawater,

which is of natural origin. A long-term change in the chemical composition of seawater cannot be detected by this way as it will be overwritten by the recalibrations with actual standard seawater.

The density is sensitive to all components, including dissolved salts and gases (and even isotopes), and can be determined without natural reference materials. If the chemical composition of standard seawater is changing in the long-term (mainly due to accumulated carbon dioxide), the density–salinity relation provides a metrological basis for detecting this change.

As possible changes in the seawater density are expected to be of the order of measurement uncertainty or even smaller, a periodic assessment should be ensured over several decades. Since the introduction of the salinity determination using standard seawater, forty years have passed without this. We propose a density measurement of any freshly prepared standard seawater batch. A well-known example of such long-term assessment is the Keeling curve of the $CO_2$ content in the atmosphere (Scripps). For standard seawater, there should be a "Keeling curve" for density in future.

**Data availability**

The complete data used to develop the density–salinity relation is provided in a digital supplement. Note that uncertainties are given either as combined uncertainties, $u$, with corresponding degrees of freedom, or as uncertainties for a probability of 95.45 %, $U$, in accordance with the Guide to the Expression of Uncertainty in Measurement (JCGM GUM, 2008).

**Appendix A: Reference water density**

The calculation of the reference densities $\rho_\text{ref}^{H_2O}$ assigned to the water reference for the substitution measurements is based on the equation of state (EOS) given by Wagner and Pruß (2002), which was adopted by the International Association of the Properties of Water and Steam in 1995 as IAPWS-95:

$$\rho_0^{H_2O} = \rho_\text{IAPWS-95},\tag{A.1}$$

where $\rho_\text{IAPWS-95}$ is valid for degassed water with VSMOW (IAEA, 2009) isotopic composition. The values calculated with this

equation were therefore corrected to the air saturation and isotopic composition of our water reference to calculate its density accurately. The equation to correct for isotopic composition was taken from Tanaka et al. (2001) and is:

$$\frac{\Delta\rho_c^{H_2O}(T_{\rho_\text{max}},p_0)}{\text{g m}^{-3}} = 0.233 \cdot \frac{\delta_{18}}{‰} + 0.0166 \cdot \frac{\delta_D}{‰},\tag{A.2}$$

where $\Delta\rho_c^{H_2O}$ is the density difference due to isotopic composition, $\delta_D$ and $\delta_{18}$ are the isotopic abundances of deuterium and oxygen-18 relative to VSMOW composition, $T_{\rho_\text{max}} = 3.98$ °C (at maximum density), and $p_0 = 101325$ Pa.

The correction for air saturation was taken from Harvey et al. (2005) and is (valid for 0 °C to 50 °C and 101325 Pa):

$$\frac{\Delta\rho_a^{H_2O}(T,p_0)}{\text{g m}^{-3}} = 0.103 - 2.371 \times 10^5 \cdot \left(\frac{T}{\text{°C}} + 75\right)^{-2.5} + 1.82 \times 10^{-7} \cdot \left(\frac{T}{\text{°C}} + 75\right)^3,\tag{A.3}$$

where $\Delta\rho_a^{H_2O}$ is the density difference due to air saturation and $T$ is the temperature.

We assumed the corrections for isotopic composition and air saturation are dependent on temperature and pressure and applied corrections in the following manner:





$$\Delta\rho_c^{H_2O}(T,p) = \frac{\Delta\rho_0^{H_2O}(T,p)}{\Delta\rho_0^{H_2O}(T_{\rho_{max}},p_0)} \cdot \Delta\rho_c^{H_2O}(T_{\rho_{max}},p_0) \text{ and} \tag{A.4}$$

$$\Delta\rho_a^{H_2O}(T,p) = \frac{\Delta\rho_0^{H_2O}(T,p)}{\Delta\rho_0^{H_2O}(T,p_0)} \cdot \Delta\rho_a^{H_2O}(T,p_0), \tag{A.5}$$

where the corrections are scaled to the density of water with VSMOW isotopic composition based on their valid states of temperature $T$ and pressure $p$. The water reference density is consequently given by:

$$\rho_{ref}^{H_2O} = \rho_0^{H_2O} + \Delta\rho_c^{H_2O} + \Delta\rho_a^{H_2O}. \tag{A.6}$$

Relative standard uncertainties for the calculated densities $\rho_{IAPWS-95}$ relevant for the temperature range of 5 °C to 35 °C given by Wagner and Pruß are $0.5 \times 10^{-6}$ for atmospheric pressure, $5 \times 10^{-6}$ for pressures up to 10 MPa, and $15 \times 10^{-6}$ for pressures up to 100 MPa. The uncertainties of corrections for isotopic composition and air saturation including the measurements and calculations contribute 10 % to the overall uncertainty in the seawater density measurements at atmospheric
pressure (Schmidt et al., 2016).

**Appendix B: Relation uncertainty**

The density–salinity relation is an empirical thermophysical equation of state, the formulation of which is determined by the underlying measurement values and their associated uncertainties, which were determined in accordance with the Guide to the Expression of Uncertainty in Measurement (GUM) adopted by the Joint Committee for Guides in Metrology (JCGM) in 2008
(JCGM GUM, 2008).

To calculate the uncertainty in predicted results for the density–salinity relation, the Monte Carlo method (MCM) as described in Supplement 2 to the GUM (JCGM GUM S2, 2011) was applied. In the MCM, $n = 15{,}000$ random values (for atmospheric and high pressures) of each particular measurement value were generated based on the associated uncertainty distribution; in case of the relative density values $\Delta\rho_0$ and $\Delta\Delta\rho_0$, this is a $t$-distribution. The result is a data set with $n$ subsets that are used to
fit the equation coefficients $n$ times. The final value of a coefficient is obtained by calculating the mean value of all (random) coefficient values resulting from the $n$ fits. The standard uncertainty in a coefficient is obtained by calculating the standard deviation. For calculation of the uncertainty in a predicted value, the correlations between the fit coefficients have to be taken into account. These are obtained by calculating the particular empirical correlation coefficients using the random data.

Since the applicability of MCM described in the GUM S2 is by definition limited to measurement models that usually involve
the use of physical laws, the uncertainty in a predicted value determined in this way may not be consistent. For this reason, the consistency of the predicted uncertainties has to be evaluated.

A common approach to evaluate the consistency of a fit equation is to compare the values of the fit residual $\Delta$ against their associated uncertainty $U(\Delta)$. A particular $U(\Delta)$ is calculated using the law of propagation of uncertainty:

$$U(\Delta) = \sqrt{U(\Delta\rho_m)^2 + U(\Delta\rho_p)^2 + 2 \cdot U(\Delta\rho_m) \cdot U(\Delta\rho_p) \cdot r(\Delta\rho_m, \Delta\rho_p)}, \tag{B.1}$$

where $r(\Delta\rho_m, \Delta\rho_p)$ is the empirical correlation coefficient of the predicted and the measured value. Because in the fit process e.g. the residual sum of squares (RSS) is minimized, the predicted and measured densities are necessarily correlated; hence, $r(\Delta\rho_m, \Delta\rho_p) \neq 0$. Since this is commonly not considered in the consistency verification, the uncertainty in predicted values may be over- or underestimated. The correlation coefficient $r(\Delta\rho_m, \Delta\rho_p) \neq 0$ was obtained by calculating the predicted value $n$ times using the $n$ subsets of the fit coefficients gained with the MCM described above.



Next, every calculated residual uncertainty at a probability of 95.45 % was compared to the corresponding residual to evaluate the uncertainty, which is associated to the predicted value. In case of the density–salinity relation consistency verification, 95.45 % of the residuals had to be smaller than their associated uncertainties, thus $|\Delta| = |\Delta\rho_\mathrm{m} - \Delta\rho_\mathrm{p}| \leq U(\Delta)$. When this was not the case and more than 4.55 % of the residuals were higher than their corresponding uncertainties, the uncertainty in the predicted value was increased gradually until the criterion was fulfilled. The increased uncertainty was then adopted for any predicted value in the corresponding atmospheric or high-pressure region.

**Acknowledgements**

This work was funded by the European Metrology Research Programme, EMRP Project ENV05. The EMRP is jointly funded by the EMRP participating countries within EURAMET and the European Union.

The authors greatly value the silicate concentration measurements obtained from seawater performed by Kai-Uwe Ludwichowski in the *Alfred-Wegener-Institut für Polar- und Meeresforschung* (AWI) and would like to thank Dr. Gereon Budéus (AWI) for valuable discussions on oceanographic matters.

This article contributes to the tasks of the Joint SCOR/IAPWS/IAPSO Committee on the Properties of Seawater (JCS).

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

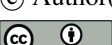



**Table 1.** Summary of the batches of the standard seawater samples.

| Date of manufacture mm/yyyy | Salinity | | | Homogeneity | | Calibration | | |
|---|---|---|---|---|---|---|---|---|
| | $S$ | $U$ [a] | $\nu_{\text{eff}}$ [d] | $\sigma$ [b] | $\nu$ [e] | $\sigma$ [c] | $\nu$ [e] | Reference |
| 10/2011 | 4.9958 | 0.0006 | 4 | 0.0000 | 4 | 0.0002 | 4 | P154 |
| 03/2011 | 9.9887 | 0.0006 | 6 | 0.0001 [f] | 4 [f] | 0.0002 [f] | 4 [f] | P153 |
| 10/2011 | 14.9999 | 0.0005 | 8 | 0.0001 | 4 | 0.0002 | 4 | P154 |
| 10/2011 | 20.0009 | 0.0007 | 7 | 0.0001 | 4 | 0.0002 | 4 | P154 |
| 10/2011 | 25.0047 | 0.0005 | 17 | 0.0001 | 4 | 0.0002 | 4 | P154 |
| 03/2011 | 29.9689 | 0.0006 | 25 | 0.0001 [f] | 4 [f] | 0.0002 [f] | 4 [f] | P153 |
| 03/2011 | 34.9917 | 0.0004 | ∞ | – | – | – | – | P153 |

[a] Uncertainty calculated based on [b], [c], and reference salinity (of standard seawater).
[b] Mean standard deviation of 5 samples from batches delivered.
[c] Mean standard deviation of 5 samples used for calibration.
[d] Effective degrees of freedom calculated based on those of homogeneity, calibration, and reference salinity.
[e] Degrees of freedom.
[f] Values are estimated.

**Table 2.** Isotopic abundances of water and seawater (NSW – natural, DSW – diluted).

| Type | $S$ | $\delta_{\text{D}}$ | $\sigma$ | $\delta_{18}$ | $\sigma$ | Source |
|---|---|---|---|---|---|---|
| – | – | ‰ | ‰ | ‰ | ‰ | – |
| NSW | 36.4 | 6.8 | 1.0 [a] | 1.06 | 0.10 [a] | (Ostlund et al., 1987) |
| $H_2O$ | – | –40.0 | 1.0 | –6.50 | 0.10 | (Darling et al., 2003) |
| DSW | 5 | –33.8 | 0.9 | –5.50 | 0.09 | – |
| DSW | 10 | –27.5 | 0.8 | –4.48 | 0.08 | – |
| DSW | 15 | –21.1 | 0.7 | –3.45 | 0.07 | – |
| DSW | 20 | –14.7 | 0.7 | –2.42 | 0.07 | – |
| DSW | 25 | –8.2 | 0.8 | –1.37 | 0.08 | – |
| DSW | 30 | –1.7 | 0.8 | –0.32 | 0.08 | – |
| IAPSO SSW | 35 | 4.9 | 1.0 | 0.76 | 0.10 | – |

[a] Value is estimated.

**Table 3.** Dissolved silicate molality of some DSW samples.

| Vessel | Salinity | Storage time in years | Silicate μmol kg⁻¹ | Uncertainty μmol kg⁻¹ | Batch |
|---|---|---|---|---|---|
| 1 | 5 | 4.1 | 36.1 | 5.4 | P154 |
| 1 | 10 | 4.7 | 43.2 | 7.2 | P153 |
| 2 | 10 | 4.7 | 48.5 | | P153 |
| 1 | 15 | 4.1 | 37.9 | 5.6 | P154 |
| 1 | 20 | 4.1 | 41.4 | 4.0 | P154 |
| 2 | 20 | 4.1 | 39.5 | | P154 |
| 1 | 25 | 4.1 | 39.7 | 4.0 | P154 |
| 1 | 30 | 4.7 | 57.6 | 6.0 | P153 |
| 2 | 30 | 4.7 | 59.9 | | P153 |
| - | 35 | 4.7 | 61.3 [a] | 6.2 [a] | P153 |

[a] Estimated based on silicate molalities for salinities of 10 and 30.





**Table 4.** Values of the coefficients $a_{i,j}$.

| i | j | Value | | i | j | Value | | i | j | Value | |
|---|---|---|---|---|---|---|---|---|---|---|---|
| 0 | 0 | 2.65627133 | $\times 10^{+2}$ | 1 | 1 | 8.0658117 | $\times 10^{+1}$ | 2 | 3 | −4.1658 | $\times 10^{-1}$ |
| 0 | 1 | −2.272462 | $\times 10^{+1}$ | 1 | 2 | −8.62107 | $\times 10^{0}$ | 3 | 0 | −1.996354156 | $\times 10^{+3}$ |
| 0 | 2 | 3.17932 | $\times 10^{0}$ | 1 | 3 | 6.3513 | $\times 10^{-1}$ | 3 | 1 | 6.332479 | $\times 10^{+1}$ |
| 0 | 3 | −2.78076 | $\times 10^{-1}$ | 1 | 4 | 6.7777 | $\times 10^{-2}$ | 3 | 2 | −2.182108 | $\times 10^{0}$ |
| 0 | 4 | −3.7051 | $\times 10^{-2}$ | 2 | 0 | 2.182680018 | $\times 10^{+3}$ | 4 | 0 | 9.16301655 | $\times 10^{+2}$ |
| 0 | 5 | −6.648 | $\times 10^{-3}$ | 2 | 1 | −1.0724787 | $\times 10^{+2}$ | 4 | 1 | −1.4043174 | $\times 10^{+1}$ |
| 1 | 0 | −1.198640497 | $\times 10^{+3}$ | 2 | 2 | 7.686316 | $\times 10^{0}$ | 5 | 0 | −1.68713114 | $\times 10^{+2}$ |

**Table 5.** Values of the coefficients $b_{i,j,k}$.

| i | j | k | Value | | i | j | k | Value | |
|---|---|---|---|---|---|---|---|---|---|
| 0 | 0 | 0 | −7.739482 | $\times 10^{+2}$ | 1 | 0 | 3 | 3.8065 | $\times 10^{-1}$ |
| 0 | 0 | 1 | 7.621224 | $\times 10^{+1}$ | 1 | 1 | 0 | 2.09786 | $\times 10^{0}$ |
| 0 | 0 | 2 | −2.47174 | $\times 10^{0}$ | 1 | 1 | 1 | 4.38047 | $\times 10^{0}$ |
| 0 | 0 | 3 | −5.109 | $\times 10^{-1}$ | 1 | 1 | 2 | −2.5183 | $\times 10^{-1}$ |
| 0 | 0 | 4 | 5.975 | $\times 10^{-2}$ | 1 | 2 | 0 | 8.72384 | $\times 10^{0}$ |
| 0 | 1 | 0 | 2.95926 | $\times 10^{0}$ | 1 | 2 | 1 | 1.7845 | $\times 10^{0}$ |
| 0 | 1 | 1 | −1.98326 | $\times 10^{0}$ | 1 | 3 | 0 | −1.2344 | $\times 10^{-1}$ |
| 0 | 1 | 2 | 5.0082 | $\times 10^{-1}$ | 2 | 0 | 0 | −3.72241428 | $\times 10^{+3}$ |
| 0 | 1 | 3 | −6.353 | $\times 10^{-2}$ | 2 | 0 | 1 | 1.8587744 | $\times 10^{+2}$ |
| 0 | 2 | 0 | −4.73032 | $\times 10^{0}$ | 2 | 0 | 2 | −2.80757 | $\times 10^{0}$ |
| 0 | 2 | 1 | −1.2834 | $\times 10^{0}$ | 2 | 1 | 0 | −1.147437 | $\times 10^{+1}$ |
| 0 | 2 | 2 | −7.863 | $\times 10^{-2}$ | 2 | 1 | 1 | −2.9345 | $\times 10^{0}$ |
| 0 | 3 | 0 | 4.9266 | $\times 10^{-1}$ | 2 | 2 | 0 | −4.66432 | $\times 10^{0}$ |
| 0 | 3 | 1 | −1.9762 | $\times 10^{-1}$ | 3 | 0 | 0 | 2.2414666 | $\times 10^{+3}$ |
| 0 | 4 | 0 | −5.466 | $\times 10^{-2}$ | 3 | 0 | 1 | −5.56069 | $\times 10^{+1}$ |
| 1 | 0 | 0 | 2.7623136 | $\times 10^{+3}$ | 3 | 1 | 0 | 6.98502 | $\times 10^{0}$ |
| 1 | 0 | 1 | −2.061301 | $\times 10^{+2}$ | 4 | 0 | 0 | −5.0878713 | $\times 10^{-6}$ |
| 1 | 0 | 2 | 5.30055 | $\times 10^{0}$ | | | | | |

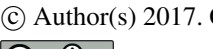
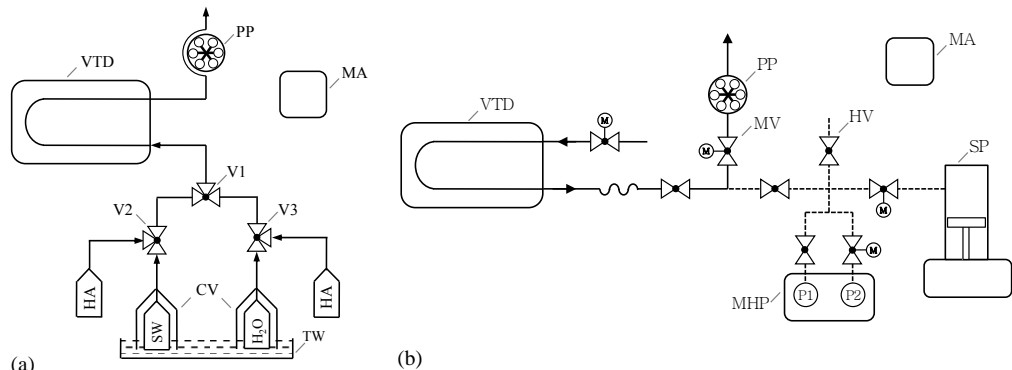

| | | |
|---|---|---|
| (a) | | (b) |

**Figure 1.** Set–up used to measure the seawater density (a) at atmospheric pressure and (b) at high pressures (Schmidt et al.,
2016). The arrows indicate flow direction in capillary tubes.

VTD – Densimeter, PP – Peristaltic pump, V1 – Liquid switching valve, V2/V3 – Air switching valves, SW – Seawater,

5 H₂O – Water, HA – Humid air, CV – Cover, TW – Tap water, MA – Manometer for atmospheric pressure, MV – Motor-
driven valve, HV – Manual valve, SP – Syringe pump, MHP – Manometer for high pressure (P1 – Full-range sensor, P2 –
Low-range sensor). Dashed lines indicate tubes filled with oil.

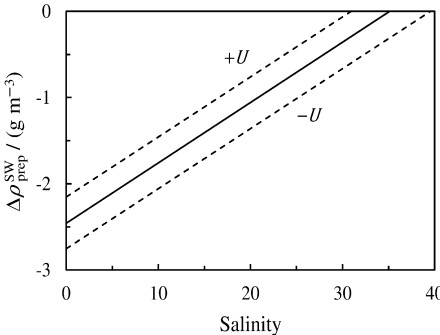

**Figure 2.** Density difference $\Delta\rho_{\mathrm{prep}}^{\mathrm{SW}}$ caused by isotopic water composition change (relative to IAPSO SSW) during preparation.

$U$ – Estimated uncertainty.





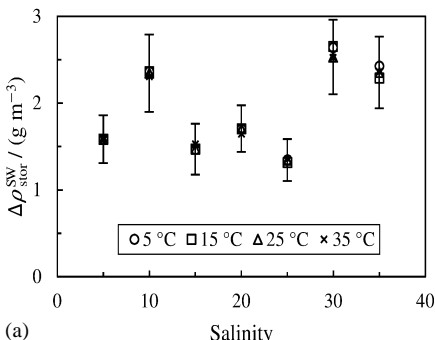
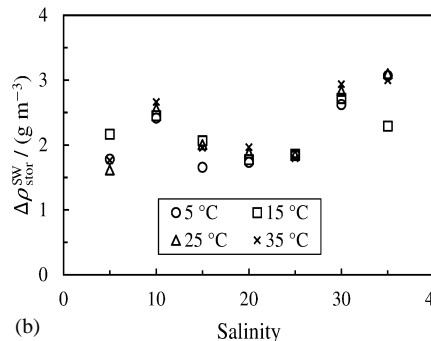

(a)          Salinity          (b)          Salinity

**Figure 3.** Seawater density increase $\Delta\rho_{stor}^{SW}$ caused by dissolution of glass material during storage. Some calculated values of samples used for density measurements (a) at atmospheric pressure and (b) at high pressures (b). Uncertainty bars in (a) are examples that indicate some uncertainties assigned to values at 25 °C.

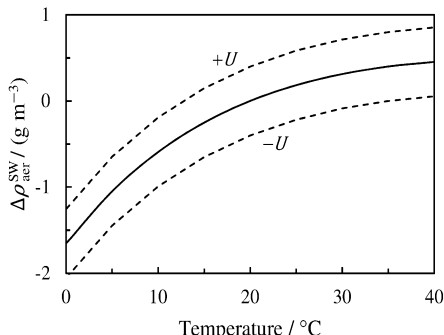

**Figure 4.** Density correction due to air saturation correction based on 100 % saturation at 20 °C. $U$ – Estimated uncertainty.

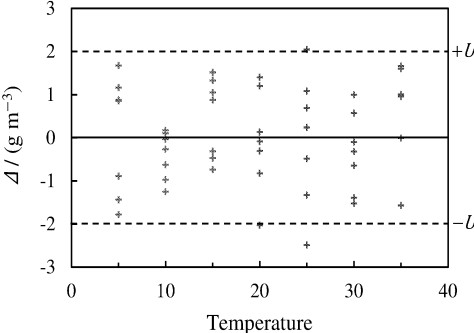

10  **Figure 5.** Residuals $\Delta$ (measured minus the predicted values) resulting from the fit of $\Delta\rho_0^{SW}(p_0)$. $U$ – Uncertainty in the density–salinity relation.



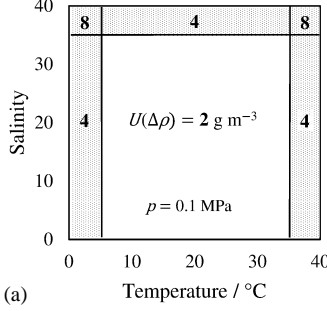
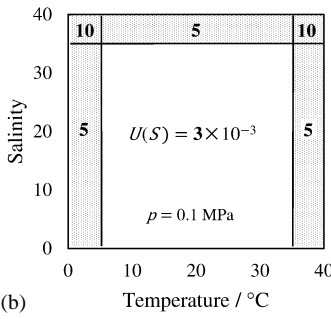

(a)     (b)

**Figure 6.** Uncertainty in the density–salinity relation at 101325 Pa. (a) Uncertainty in the relative density of air-saturated seawater, $U(\Delta\rho^{SW})$, that results from a calculation using salinity and temperature values. (b) Uncertainty in salinity, $U(S)$, that results from an inverse calculation using the relative density of air-saturated seawater and temperature values. The white area indicates the measurement region equal to that of the data set used for fitting. The grey area indicates the extrapolation region.

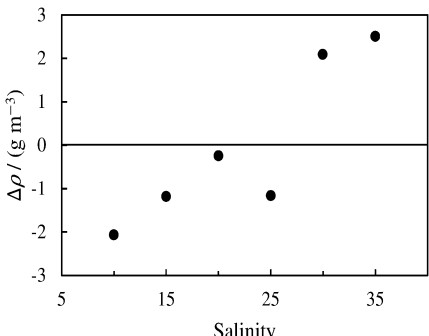

**Figure 7.** Deviation of measured from predicted seawater densities $\Delta\rho$ in the extrapolation region at 1 °C and atmospheric pressure.





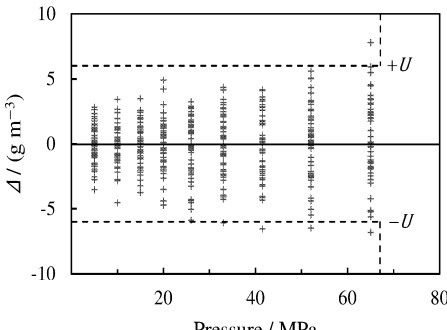

**Figure 8.** Residuals $\Delta$ (measured minus the predicted values) yielded by the fit of $\Delta\Delta\rho_0^{\mathrm{SW}}(p - p_0)$. $U$ – Uncertainty in the density–salinity relation for high pressures.

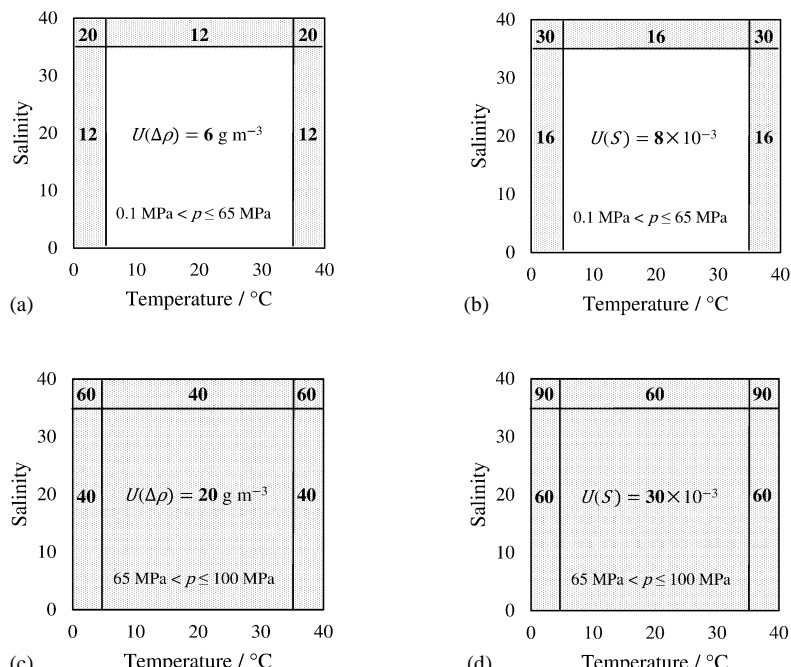

**Figure 9.** Uncertainty in the density–salinity relation at high pressures. Uncertainty in the relative density of air-saturated seawater, $U(\Delta\rho^{\mathrm{SW}})$, that results from a calculation using salinity and temperature values for pressures (a) up to 65 MPa and (c) up to 100 MPa. Uncertainty in salinity, $U(S)$, that results from an inverse calculation using the relative density of air-saturated seawater and temperature values for pressures (b) up to 65 MPa and (d) up to 100 MPa. The white area indicates the

10 interpolation region equal to that of the data set used for fitting. The grey area indicates the extrapolation region.



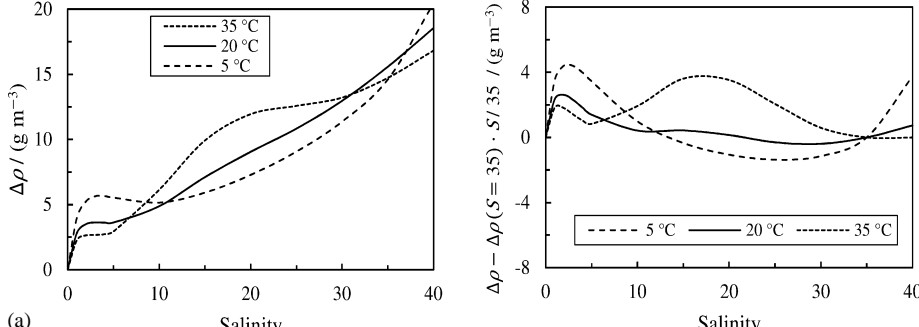

**Figure 10.** Density deviation of TEOS-10 from the density–salinity relation (i.e. TEOS-10 minus DSR) for degassed seawater at selected temperatures and atmospheric pressure. (a) The deviation $\Delta\rho$ increases linearly with salinity. The uncertainty in the deviation is 8 g m$^{-3}$, and is significantly exceeded at salinities higher than 20. (b) By contrast, the salinity-35-reduced deviation is less than 5 g m$^{-3}$.

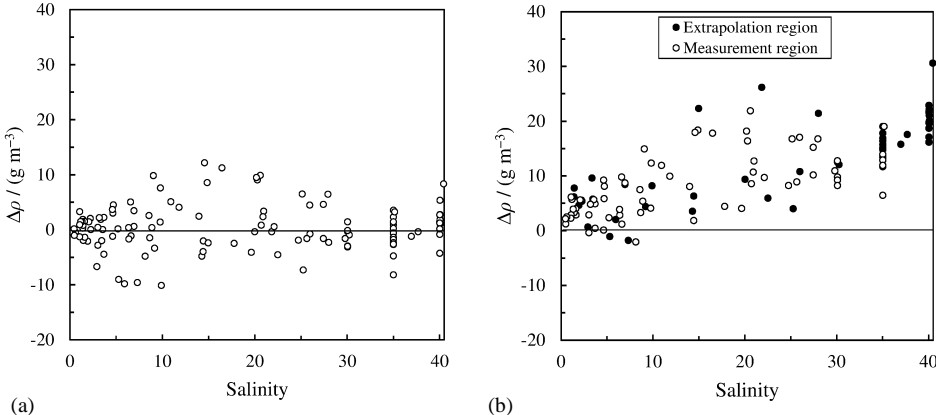

**Figure 11.** Deviation of densities obtained from standard seawater using a magnetic float densimeter by Millero et al. (JPOTS, 1981c, pp. 51–56). (a) The deviations from TEOS-10 (i.e. Millero et al. minus TEOS-10) suggest that TEOS-10 is well fitted to the densities of Millero et al. (b) The deviations from the density–salinity relation (i.e. Millero et al. minus DSR) suggest a deviation that systematically depends on salinity. The density uncertainty calculated using the accuracy and reproducibility claimed by Millero et al. is 2 g m$^{-3}$ and is significantly exceeded by most of the deviations.




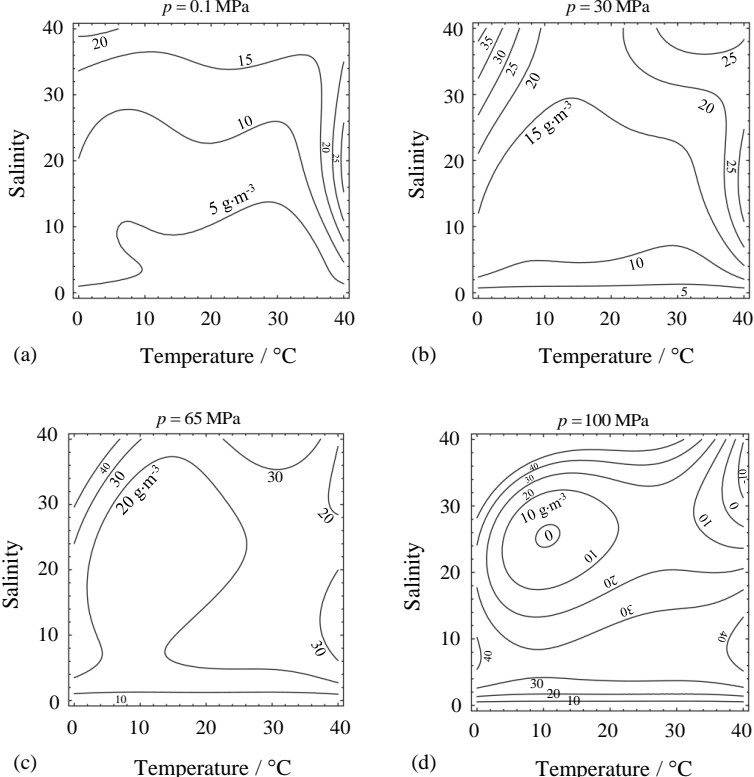

**Figure 12.** Density deviation of TEOS-10 from the density–salinity relation (i.e. TEOS-10 minus DSR) for degassed seawater (a) at atmospheric pressure, (b) at 30 MPa, (c) at 65 MPa, and (d) at 100 MPa. The uncertainties in deviations are 8 g m⁻³, 26 g m⁻³, 26 g m⁻³, and 33 g m⁻³. The deviations only exceed these uncertainties significantly at atmospheric pressure.