# Peer review of "The density-salinity relation of standard seawater"

_Ocean Science, 2017_

## Referee Comment (RC1) · Anonymous Referee #1 · 12 Sep 2017

Review: Ocean Science: Schmidt et al., The density-salinity relation of standard seawater

**General Comments:**

This work is a good contribution toward addressing some of the challenges expressed in the cited 2016 *Metrologia* paper by Pawlowicz et al. The idea of using density as a measure of salinity, and/or as a measure of changing seawater composition, is attractive. But there are many difficult details, especially because the measurement uncertainty needed for this to be successful is of similar magnitude to several small corrections (such as air saturation and isotopic composition).

This work combines careful density measurements on standard seawater with careful analysis of small corrections to develop a relationship between density and salinity. At least in principle, deviations between this relationship and salinity determined by conductivity could give an indication of changes in seawater composition. Other valuable contributions of this work include the data themselves, which are probably more accurate and better documented than any previous measurements of seawater density, the analysis for how dilution of samples with fresh water of different isotopic composition can affect the density, and the apparent finding that the density data most important in fitting TEOS-10 in this region were not as accurate as has been assumed (and may have a systematic deviation).

Overall, I recommend publication with minor revisions. Some scientific comments and comments about the clarity of the writing follow.

**Specific Comments**

- 1. The isotopic composition of their reference water is given at the top of page 5, but it is not stated whether that is measured before or after degassing. This could be important, since degassing by boiling could change the isotopic composition (if more than a few percent is boiled off).
- 2. On page 7, the use of the survey of Darling to estimate the isotopic composition of dilution water for SSW. Is it known how the dilution water was purified? If it was purified by distillation, that would change the isotopic composition. Of course it would be even better if the actual tap water used by the supplier could be analyzed, but the authors do not control that.
- 3. On page 7, the description of VSMOW preparation is incorrect; melted ice is not involved (the ice is used for the related SLAP standard).
- 4. I like the suggestion that density should be measured on each new seawater batch. Are there other measurements that should be done at that time in order to give information that might be useful (perhaps even useful decades later)? I would think isotopic composition might be such a measurement. Maybe silicate concentration? Are there other key ions where knowledge of concentration would be useful?
- 5. I wonder about the unusual slopes at low salinity in Fig. 10. These suggest that the initial variation of density with salinity follows a different function in TEOS-10 compared to Eq. (13). Is there more the authors can say about this? Is the correct behavior known from theory? Intuitively I would think the initial dependence should be linear, but my intuition may not be correct (some things involving electrolytes have square-root terms in their concentration dependence).

**Technical Comments**

- 1. Some of the writing is unclear what the authors mean by "composition." It took me some time to figure out that the authors do not mean change in the salinity itself (which would qualify as a composition change in normal chemical terminology), but only a change in the relative proportion of dissolved species. I do not have a specific suggestion for making this more clear, but the authors should look for some way to do so. Perhaps there should be a footnote similar to the one for "salinity."
- 2. Page 2, line 11, instead of "silicon" it should probably be "silicate".
- 3. Page 10, line 11, I don't think "gassing" is a correct English word here.
- 4. Page 12, line 25, "for safety reasons" makes it sound like they were worried about an explosion or something. Probably should say something like "to be certain".
- 5. Page 16, line 16, "cylindring" is not an English word; it probably should be "cylindrical".
- 6. Some of the writing on page 16 about examining possible sources of error in Millero's magnetic flotation densimetry is unclear. It is not clear whether they are describing checks that were done by Millero in the paper being discussed, or checks that the authors of the current paper are making for the first time. I think it is probably the latter, in which case it would greatly help the clarity if, for example, line 15 said "we carried out a representative calculation".
- 7. In Tables 3 and 4, it would be helpful to the reader if the caption stated the number of the equation to which the coefficients correspond. This could be simply be done by adding "in Eq. (X)" at the end in both cases.

---

## Author Comment (AC1) · 19 Sep 2017

GENERAL COMMMENTS

Thanks for the comments and questions, whose clarification contributes to the improvement of our article.

It is true that a certain amount of know-how and care is required to determine the salinity by density measurement with high accuracy. However, this know-how has been made available in this and our previous publication on our substitution method. Furthermore, it was shown in density comparison measurements, wherein oceanographic institutes were also involved, that the measuring equipment and time expenditure for a highly accurate density measurement at atmospheric pressure (in the lab) is moderate

provided that the know-how is consistently applied.

SPECIFIC AND TECHNICAL COMMENTS, ADDITIONAL INFORMATION AND COMMENTS

Open the link provided below.

Please also note the supplement to this comment:
https://www.ocean-sci-discuss.net/os-2017-71/os-2017-71-AC1-supplement.pdf

[Figure]

**Supplement:**

**Response to RC1 (12 Sep 2017)**

**General comments**

Thanks for the comments and questions, whose clarification contributes to the improvement of our article.

It is true that a certain amount of know-how and care is required to determine the salinity by density measurement with high accuracy. However, this know-how has been made available in this and our previous publication on our substitution method. Furthermore, it was shown in density comparison measurements, wherein oceanographic institutes were also involved, that the measuring equipment and time expenditure for a highly accurate density measurement at atmospheric pressure (in the lab) is moderate provided that the know-how is consistently applied [1].

**Specific comments**

**Page 4 Line 1**. Was the isotopic composition measured before or after degassing?

The following change was made to the manuscript (from <left> to <right>):

| | |
|---|---|
| The isotopic abundances of deuterium and of oxygen-18 against Vienna Standard Mean Ocean Water were −59 ‰ and −8.5 ‰. Details on the determination of the air saturation and isotopic composition were given by Schmidt et al. (2016). | The isotopic abundances of deuterium and of oxygen-18 against Vienna Standard Mean Ocean Water were −59 ‰ and −8.5 ‰. The abundances were measured before and after degassing and no significant differences were found. Details on the determination of the air saturation and isotopic composition were given by Schmidt et al. (2016). |

**P4 L1.** What amount of water can be evaporated without a significant change in the water density caused by heavy isotope enrichment?

The change in density of water caused by boiling is shown in the figure below that is based on the theoretical model described at the end. If 10 % water is evaporated, then the density increases by a maximum of 0.3 g m$^{-3}$ suggesting that moderate boiling insignificantly enriches heavy isotopes in terms of density.

**Fig.** Density increase of water, $\Delta\rho$, due to isotope enrichment during boiling. Calculated curves for evaporation into a pure water vapour phase at 20 ℃ (- - -), 60 ℃ (——) und 100 ℃.

[Figure]

**P7 L16−9.** How was the water purified that was used to prepare the seawater samples?

[1] Schmidt, H and Wolf, H.: Results of the Seawater Density Comparison 2014, IAPWS Annual Meeting 2015, 29th June to 3rd July 2015, Stockholm, Sweden, doi:10.13140/RG.2.2.28647.55207.

For the preparation of standard seawater (with S = 35), seawater from the North Atlantic Ocean (with S > 35) was diluted with deionized water[2]. This water is ultrapurified in a two stage process including, ia, reverse osmosis (RO), deionization (DI), UV, ion exchange, and microfiltration [3]. Such deionized water was also used for the preparation of diluted standard seawater (with S < 35) [4].

We know from own investigations of our tap water and deionized water therefrom that the isotopic composition is not changed by the ultrapurification process.

The following change was made to the manuscript (from <left> to <right>):

| | |
|---|---|
| The water, which is  and used for dilution of the natural seawater, is tap water from Havant, UK, where the supplier of the IAPSO SSW is located. | The water, which is deionized and used for dilution of the natural seawater (N. Higgs, personal communication, 2011), is tap water from Havant, UK, where the supplier of the IAPSO SSW is located. |

**P7 L5−6.** What was VSMOW prepared from?

VSMOW was, of course, not prepared from melted ice.

The following change was made to the manuscript (from <left> to <right>):

| | |
|---|---|
| The isotopic abundance of a water sample is usually stated relative to that of the reference material VSMOW, whose isotopic composition is based on a mixture of ocean waters  (IAEA, 2006). | The isotopic abundance of a water sample is usually stated relative to that of the reference material VSMOW, whose isotopic composition is based on a mixture of ocean waters (IAEA, 2006). |

**P19 L12−3.** Which regular measurements obtained from standard seawater should be carried out in addition to density?

**Dissolved silicate**

It was helpful that there were published measurements of silicate in standard seawater from 1978, as these made it possible to check whether the standard seawater used for the density measurements of TEOS-10 had the same silicate content as the seawater used for our density measurements. Small amounts of dissolved silicate do not affect the conductivity significantly, but the density.

**Carbon dioxide**

As pointed out in the introduction of our article, $CO_2$ enrichment in the atmosphere will lead to $CO_2$ absorption into the oceans. The $CO_2$ content of standard seawater is promptly affected, as standard seawater is sampled close to the ocean surface and equilibrates with the atmosphere during its production over several weeks in an open vessel [2]. The change of the absolute salinity due do $CO_2$ absorption was modelled; it is expected that the absolute salinity will change by 0.005 until the year 2100 taking the year 2000 as reference [5]. Since part of the $CO_2$ contributes to the seawater conductivity, the (practical) salinity is also affected. The effect on the density–(practical) salinity relation has not been investigated, but probably should before there are any regular measurements of $CO_2$ absorbed in standard seawater.

[2] Bacon, S., Culkin, F., Higgs, N., and Ridout, P.: IAPSO standard seawater: definition of the uncertainty in the calibration procedure and stability of recent batches, Journal of Atmospheric and Oceanic Technology, 24, 1785–1799, doi:10.1175/JTECH2081.1, 2007.
[3] R. Williams (Ocean Scientific International Limited), personal communication, 2017.
[4] N. Higgs (Ocean Scientific International Limited), personal communication, 2011.
[5] Woosley, R. J., Huang, F., and Millero, F. J.: Estimating absolute salinity (SA) in the world's oceans using density and composition, Deep-Sea Research I, 93, 14–20, doi:10.1016/J.DSR.2014.07.009, 2014.

If $CO_2$ caused inconsistencies in the density–(practical) salinity relation, regular density measurements would reveal.

**Isotopic composition**

The isotopic composition of near-surface seawater is usually determined by the ratio of evaporation and rainfall. The isotopic composition therefore correlates locally with salinity and temporally with season. However, taking the continuous "calving" of Greenland ice glaciers into account, which have a significantly different isotopic composition compared to seawater (as SLAP vs. VSMOW?), there might be a long-term change in the seawater isotopic composition. If this drift is measurable in standard seawater is a question of uncertainty in measurement (as it would be for measurable density changes) and mixing due to oceanic currents in terms of quantity, whereof we (as non-oceanographers) have insufficient knowledge to answer adequately.

**Unkown components**

A density measurement can detect a change in the seawater and is performed quickly. By contrast, a chemical analysis can identify changes, but the effort is very high. So why not combining them? If a change in density is detected, a complete (or targeted) chemical analysis identifies the cause.

**P14 L34–5/P31 Fig. 10.** How does the mathematical formulation of TEOS-10-density differ from that of the density–salinity relation, as this seems to cause unexpectedly large density deviations for low salinities?

In developing the density–salinity relation several mathematical formulations were considered including those with (salinity-)square-root terms, as these were used for TEOS-10 and EOS-80. However, such functions did not represent the measurement values as well as the linear terms used for the density–salinity relation.

**Page 1 Lines 6 to 7, P1 L25–27, P19 L5–6.** It should be made clearer what is meant by 'composition' and 'composition change'.

The following changes were made to the manuscript (from <left> to <right>):

| | |
|---|---|
| The determination of salinity by means of electrical conductivity relies on  in the North Atlantic Ocean, as standard seawater, which is required for salinometer calibration, is produced therefrom. | The determination of salinity by means of electrical conductivity relies on constant relative salt proportions in the North Atlantic Ocean, as standard seawater, which is required for salinometer calibration, is produced therefrom. |
| An unconditional prerequisite for the comparability of salinity measurements over long periods is, therefore, . | An unconditional prerequisite for the comparability of salinity measurements over long periods is, therefore, that the relative salt proportions in standard seawater are stable. |
| A long-term change in the  seawater cannot be detected by this way as it will be overwritten by the recalibrations with actual standard seawater. | A long-term change in the relative salt proportions in seawater cannot be detected by this way as it will be overwritten by the recalibrations with actual standard seawater. |

**P2 L11.** Replace 'silicon' by 'silicate'.

The following change was made to the manuscript (from <left> to <right>):

| | |
|---|---|
| [..] during storage, glass container material dissolves in the seawater, which is mainly  [..] | [..] during storage, glass container material dissolves in the seawater, which is mainly silicate [..] |

**P10 L11.** Use a different word for "gassing".

The following change was made to the manuscript (from <left> to <right>):

| | |
|---|---|
| This temperature-dependent  is significant compared to the density measurement uncertainty. | This temperature-dependent aeration is significant compared to the density measurement uncertainty. |

**P12 L25.** Replace 'for safety reasons' by 'to be certain'.

The following change was made to the manuscript (from <left> to <right>):

| | |
|---|---|
| Finally, the coefficients were averaged from $n = 15,000$ runs . | Finally, the coefficients were averaged from $n = 15,000$ runs to be certain. |

**P16 L16.** Replace 'cylindring' by 'cylindrical'.

The following change was made to the manuscript (from <left> to <right>):

| | |
|---|---|
| [..] theoretically, $F_{\mathrm{mag}} \propto \mu \cdot I$ for a  (and circular) ring coil [..] | [..] theoretically, $F_{\mathrm{mag}} \propto \mu \cdot I$ for a cylindrical (and circular) ring coil [..] |

**P15 L20–1, P16 L15–6, P16 L28–9, P16 L33–4.** It is not clear, who carried out the representative calculations.

The following changes were made to the manuscript (from <left> to <right>):

| | |
|---|---|
| For this reason, the magnetic flotation method used by Millero et al.  in detail for possible issues. | For this reason, we examined in detail the magnetic flotation method used by Millero et al. for possible issues. |

In order to rule out this possibility, .

A further representative calculation  with the values (i–iv).

 to estimate how significant the precise height positioning of the permanent magnet is [..]

In order to rule out this possibility, we carried out a representative calculation.

We therefore carried out a further representative calculation using the values (i–iv).

We performed a final calculation to estimate how significant the precise height positioning of the permanent magnet is [..]

**P26 Tab. 4 and 5.** It is not obvious which equation the coefficients correspond to.

The following changes were made to the manuscript (from <left> to <right>):

**Table 4.** Values of the coefficients $a_{i,j}$.

**Table 5.** Values of the coefficients $b_{i,j,k}$.

**Table 4.** Values of the coefficients $a_{i,j}$ of Eq. (13).

**Table 5.** Values of the coefficients $b_{i,j,k}$ of Eq. (14).

**Isotope fractionation**

If a vessel contains a very small vapour (v) compared to a liquid phase (l) each consisting of water, and the water contains $^1$H and $^2$H, or D, atoms and $^{16}$O und $^{18}$O atoms, then the frequency of these atoms in the liquid and gas phase is different. The frequency of an isotope of the liquid relative to the vapour phase is described by means of the isotopic fractionation factor $\alpha$, which is for deuterium:

$$\alpha_D = \frac{n_D^l/n_H^l}{n_D^v/n_H^v},$$

where $n$ is the amount-of-substance. For example, $n_D^l$ is the amount-of-substance deuterium in the liquid phase. The fractionation factor of deuterium and oxygen-18 is temperature-dependent[6].

**Isotope enrichment**

If a very small amount is repeatedly removed from the vapour phase at very long intervals, some molecules from the liquid phase „vaporize" into the vapour phase at (almost) constant temperature. The infinitesimal changes in the amounts-of-substance of H and D in the liquid and vapour phase are then linked by $dn_D^v = -dn_D^l$ and $dn_H^v = -dn_H^l$. For the ratio D to H of the vapour phase, it follows that $n_D^v/n_H^v \approx dn_D^l/dn_H^l$. Inserting this formula into above formula, transforming and integrating from state (I) at the beginning to state (II) at the end of vaporization results in:

$$\frac{n_D^{II}}{n_D^{I}} = \sqrt[\alpha_D]{\frac{n_H^{II}}{n_H^{I}}},$$

where all amount-of-substances refer to the liquid phase.

The isotopic composition of deuterium and oxygen-18 in water is given by isotopic abundances relative to VSMOW, $\delta_D$ and $\delta_{18}$, see article. The use of the isotopic abundance instead of the amount-of-substance in above formula yields:

$$\frac{\delta_D^{II} + 1}{\delta_D^{I} + 1} = \left(\frac{n_H^{II}}{n_H^{I}}\right)^{\frac{1-\alpha_D}{\alpha_D}}.$$

For oxygen-18 the formula is similar.

**Density change**

The density change due to the enrichment of deuterium and oxygen-18 can be calculated using the formula of Girard and Menaché[7] that is given in the article's appendix A.
* * *
[6] Horita, J. and Wesolowksi, D. J.: Liquid-vapor fractionation of oxygen and hydrogen isotopes of water from the freezing to the critical temperature, Geochimica et Cosmochimica Acta, 58, 16, 3425–37, doi: 10.1016/0016-7037(94)90096-5, 1994.

[7] Girard, G. and Menaché, M.: Variation de la masse volumique de l'eau en fonction de sa composition isotopique, Metrologia, 7, 83–87, doi:10.1088/0026-1394/7/3/001, 1971.

---

## Referee Comment (RC2) · Anonymous Referee #1 · 22 Sep 2017

The authors have done a good job in responding to my concerns. I just have 2 minor comments:

1) I think it would be good in the paper to briefly mention other measurements in addition to density that might be recommended to be made on standard seawater, especially measurements that might help interpret changes in density. The authors point out that silicate is in this category. Since the variations in density they are considering are of similar magnitude to variations due to isotopic composition, it seems like that would be another – especially since I believe isotopic composition measurements on water are now fairly routine due to their importance in geoscience.

2) If the proposed formulation does not obey the correct physical boundary condition at zero density, that should be mentioned. It is probably of little practical relevance, since

I think the authors do not anticipate using their relationship at salinities below 5 or 10, but a brief statement might keep other people in the future from trying to use it at dilute conditions where its behavior is not quite correct.

---

## Referee Comment (RC3) · Anonymous Referee #2 · 25 Sep 2017

Hannes Schmidt et al. present a very well documented paper about a subject which realization was waited for a few years: the measurement of the density of standard seawater bottles. This subject is of a great importance in the way to unsure a traceability on long time scales of the standard seawater used to calibrate laboratory salinometers.

The presentation and the explanations are clear and easy to follow. In each paragraph, numerous and useful details are given, but the essential of the method used is described in another publication of Metrologia 53 (2016). This work shows what metrology and metrologists can bring to chemical and physical oceanography.

I recommend the publication of this paper after the authors will have answered the few following questions or remarks:

[Figure]

-1- The DMA 5000 is calibrated with two points at atmospheric pressure: air and water. Nothing is said about the relation and the conditions used to calibrate with the air. Does the BIPM relation (A Picard, RS Davis, M Glaser and K. Fujii, 2008, 'Revised formula for the density of moist air (CIPM-2007)', Metrologia. 45, 149-155.) was used? This point is of a great importance as it determines the offset of the instrument and it is in relation with the accuracy of measurements made beyond the density of fresh water. If the manufacturer's relation was used, could you assess the impact on the accuracy of the measurements of seawater with the DMA 5000?

2.1- About the substitution method: it allows the compensation of the drifts of the instrument, but it can't warrant its linearity for measurements made beyond the density of fresh water. This element was checked in the Metrologia's publication, by comparing to an hydrostatic weighting apparatus at atmospheric pressure. Could you remember this point in this publication, as it is of a great importance to validate your data compared to Millero results?

2.2 - At high pressure, the linearity was checked and corrected with 4 substances, the less precise having the higher densities. In this publication, their standard uncertainty is given to be 25 g/m3 and you give a standard uncertainty adjustment of 19 g/m3 at 65 MPa. In this case, how can you unsure an uncertainty of 0.008 (including the linearity in the budget) in salinity at high pressure and the deviations you give in figure 12?

3.1 - About the apparatus: tubes are clamped between the VTD, the peristaltic pump and the valve. B. Laky, of the research lab of the manufacturer Anton Paar, has made a remark about the influence of tubes on the damping frequency of the DMA. Any modification of the in/output assembly of the cell, even a small movement of attached tubes, can alter the DMA's results. The oscillation frequency can be altered with any modification of the mechanical forcing by attachments. How have you taken into account this remark?

3.2 - The DMA 5000 is quite sensitive to the inclination of its support. Apart from a

low vibration stand, a level surface is mandatory for a best accuracy. It seems that an inclination of a few degrees can affect the measurements by about 2 ppm/degree. Could you say what cares have been taken to prevent this error?

3.3 - In the pressure experiments, how can you unsure that the tube is sufficiently long to avoid diffusion of the oil in the U-tube of the DMA during measurements at high pressure?

4 - Paragraph 3.2.3, line 15: replace 'to store the of seawater' by 'to store the seawater'.

5 - in the summary, line 26, you speak of a 'linear dependence' on salinity. With the deviations given in figure 11 b), it seems that a parabolic equation could fit the data as well. Could you calculate the best polynomial, give its correlation coefficient and display it on the figure 11 b), in order to prove your assumption of a linear dependence?

---

## Author Comment (AC2) · 25 Sep 2017

MINOR COMMENTS

1. 'I think it would be good in the paper to briefly mention other measurements in addition to density that might be recommended to be made on standard seawater, especially measurements that might help interpret changes in density. The authors point out that silicate is in this category. Since the variations in density they are considering are of similar magnitude to variations due to isotopic composition, it seems like that would be another – especially since I believe isotopic composition measurements on water are now fairly routine due to their importance in geoscience.'

This is a similar comment like in the first comment (RC1). We expanded that conclusion

und refer to the answer given there.

2. Since theoretical boundary conditions at 'zero density', i.e. for S→0, are not fulfilled by the mathematical formulation of the density–salinity relation (DSR), is there a risk for inconsistencies?

Please open the link provided below.

Please also note the supplement to this comment:
https://www.ocean-sci-discuss.net/os-2017-71/os-2017-71-AC2-supplement.pdf

**Supplement:**

**Response to RC2 (22 Sep 2017)**

'I think it would be good in the paper to briefly mention other measurements in addition to density that might be recommended to be made on standard seawater, especially measurements that might help interpret changes in density. The authors point out that silicate is in this category. Since the variations in density they are considering are of similar magnitude to variations due to isotopic composition, it seems like that would be another – especially since I believe isotopic composition measurements on water are now fairly routine due to their importance in geoscience.'

This is a similar comment like in the first comment (RC1). We expanded that conclusion und refer to the answer given there.

Since theoretical boundary conditions at 'zero density', i.e. $\lim_{S \to 0} \Delta \rho(S)$, are not fulfilled by the mathematical formulation of the density–salinity relation (DSR), is there a risk for inconsistencies?

The following change was made to the paragraph at the end of Sect. 4.2:

[..] Since the mathematical formulation of the density-salinity relation is empirical and does not contain any theoretical boundary conditions for infinite dilution, as e.g. implemented in TEOS-10, the question arises whether the relation correctly predicts the density for very low salinities. Additionally, no uncertainty verification in the extrapolation region is possible using the fitting data set. Therefore, additional substitution density measurements were conducted: The density of diluted standard seawater with salinity 2 measured at some temperatures and the density of some samples of the seawater used for determination of the density–salinity relation at 1 °C were measured. The seawater with salinity 2 was prepared like the seawater with salinities from 5 to 30. Unfortunately, the precision in the salinity-2-calibration was lower, so that the uncertainty in salinity is 0.0028 corresponding to an uncertainty in density of 2.2 g m$^{-3}$. The density results were corrected to the uniform isotopic water and the chemical salt compositions as well as air saturation as described in Sect. 3. The density deviations of the corrected results from the predicted values of the density–salinity relation are shown in Fig. 7. In both cases, the deviations are well within the uncertainty in the density–salinity relation. Note that for the measurements on seawater with salinity 2, even if the uncertainty in salinity is treated as an offset to all deviations, the deviation is within its uncertainty. No inconsistencies are caused by the non-compliance with theoretical boundary conditions for very low salinities and atmospheric pressure.

[Figure]

**Figure 7.** Deviation of measured from predicted seawater densities $\Delta\rho$. (a) In the interpolation range at salinity 2 and (b) in the extrapolation region at 1 °C at atmospheric pressure, respectively. $U$ – Uncertainty in the density–salinity relation.

The following paragraph was added at the end of Sect. 4.3:

[..] As pointed out above, the mathematical formulation of the density–salinity relation is empirical and does not contain any theoretical boundary conditions for infinite dilution. This is also an issue for the density at high pressures, as here the measurement uncertainty in density is higher, thereby causing more variability in the shape of the relation for very low salinities. Therefore, additional measurements were conducted on diluted standard seawater with salinity 2 for some temperatures. The samples used were obtained from the same seawater as described above in Sect. 4.2; the corrections were similar. The density deviations of the corrected results from predicted values of the density–salinity relation are shown in Fig. 10. The deviations are well within the uncertainty in the relation. No inconsistencies are caused by the non-compliance with theoretical boundary conditions for very low salinities and high pressures.

[Figure]

**Figure 10.** Deviation of measured from predicted seawater densities $\Delta\rho$ in the interpolation range at salinity 2. $U$ – Uncertainty in the density–salinity relation.

---

## Author Comment (AC3) · 27 Sep 2017

**Response to RC3 (25 Sep 2017)**

**General comments**

Thanks for the comments and questions, whose clarification contributes to the improvement of our article.

**Specific comments**

**What is the impact of a zero-drift on the accuracy of a substitution measurement on standard seawater using a water reference? (Question 1)**

The impact is negligible, provided that the densimeter is regularly quick-adjusted by the user (as it is usual in a well-managed lab).

A maximum deviation in air density of $20 \text{ g m}^{-3}$ (reference minus measured) without a deviation in water density causes a non-considered deviation in the substitution density of seawater with salinity 35 of $0.5 \text{ g m}^{-3}$. If, additionally, there is a deviation in water density of $-10 \text{ g m}^{-3}$ (in the opposite direction), then a non-considered deviation in the substitution density of $0.8 \text{ g m}^{-3}$ results.

However, we never saw such deviations during our substitution measurements. Additionally, such deviations are random, and are therefore considered in the repeatability of a substitution measurement, which for our measurements at atmospheric pressure was $1 \text{ g m}^{-3}$.

**How is a DMA 5000 M densimeter adjusted?**

The DMA (= density measurement apparatus) is adjusted by the manufacturer. The standards used for this purpose are multiple reference fluids (including air and water), whereof the density and viscosity are known, respectively. A quick-adjustment is provided to the customer using air and water. We performed a quick-adjustment before any substitution measurement.

**Which formula is used to calculate the air density by the DMA 5000 M?**

For the calculation of air density, the formula given by Spieweck and Bettin[1] for a relative humidity of 50 % is used by the internal firmware of the device[2]. The air pressure is either measured by an internal barometer or provided by the customer. We used an external high precision barometer that was calibrated to provide the air pressure. Note that the DMA 5000 M manual[2] also contains data tables.

**What is the density difference between this formula and the recent reference formula?**

The formula given by Spieweck and Bettin[1] deviates significantly less than $1 \text{ g m}^{-3}$ for 20 °C and 50 %rh from the recent formulation of air density CIPM-2007 [3]. A change in relative humidity of 10 % at 20 °C changes the air density by $1 \text{ g m}^{-3}$. A deviation in the air density of $1 \text{ g m}^{-3}$ yields the deviation in the seawater (substitution) density of $0.025 \text{ g m}^{-3}$ that is insignificant in terms of uncertainty. We therefore decided to use the internal firmware calculation.
* * *
[1] Spieweck, F. and Bettin, H.: Review – Solid and liquid density determination, Technisches Messen 7/8, 1992, p. 291.

[2] Anton Paar GmbH: Manual – DMA 4100, DMA 4500 M, DMA 5000 M, Firmware-Version: V2.20, 18th January 2012.
*A more recent manual (in English) may be obtained using the link (after registration):*
https://www.anton-paar.com/?eID=documentsDownload&document=5471&L=1

[3] Picard, A., Davis, R. S., Gläser, M. and Fujii, K.: Revised formula for the density of moist air (CIPM-2007), Metrologia, 45, 149–155, doi:10.1088/0026-1394/45/2/004, 2007.

**How can the impact be estimated?**

The impact can be estimated using the formula:

$$\Delta\rho_{SW} = (1 - \gamma) \cdot \Delta\rho_A + \gamma \cdot \Delta\rho_{H_2O} \text{ with } \gamma = (\rho_{SW} - \rho_A)/(\rho_{H_2O} - \rho_A),$$

where $\rho_A$ and $\Delta\rho_A$ are the air reference density and deviation therefrom (ref. minus meas.), $\rho_{H_2O}$ and $\Delta\rho_{H_2O}$ are the water reference density and deviation therefrom (ref. minus meas.), and $\Delta\rho_{SW}$ is the difference between substitution and measured seawater density (subs. minus meas.).

The idea behind the formula is illustrated in (and may be derived from) the figure below.

**Figure.** Linear characteristic curve of an ideal densimeter (——) and of a densimeter with a zero-offset ($-\cdot-$).

[Figure]

Are there details on the substitution method yet not provided?

The substitution method does not correct a nonlinear characteristic curve (compare the linear characteristic curve in the figure above). Since the linearity of the DMA 5000 M and HP was checked before, please provide a hint to the reference. (Hint 2.1)

The following change was made to the manuscript at Page 3 Line 28 (from left to right):

2.1 Substitution method
[..] The uncertainty of a corrected seawater density resulting from a substitution measurement mainly depends on the uncertainty of the water reference, but also on the similarity of seawater and water in terms of their relevant thermophysical properties, as well as on the stability and linear characteristics of the densimeter used.

2.2 Materials
[..]

2.1 Substitution method
[..] The uncertainty of a corrected seawater density resulting from a substitution measurement mainly depends on the uncertainty of the water reference, but also on the similarity of seawater and water in terms of their relevant thermophysical properties, as well as on the stability and linear characteristics of the densimeter used. Note that the linearity is regularly checked in measurements on reference liquids with densities between $700 \text{ kg m}^{-3}$ and $1600 \text{ kg m}^{-3}$ ; furthermore, the linearity was particularly validated in the seawater density range by means of comparison measurements against a hydrostatic weighing apparatus for both the densimeters used for atmospheric and high pressure; details have been given by Schmidt et al. (2016).

2.2 Materials
[..]

**Uncertainty at high pressure (Q. 2.2)**

**Reviewer**: 'In this publication, their standard uncertainty is given to be $25 \, \mathrm{g \, m^{-3}}$ and you give a standard uncertainty adjustment of $19 \, \mathrm{g \, m^{-3}}$ at $65 \, \mathrm{MPa}$. In this case, how can you ensure an uncertainty of 0.008 (including the linearity in the budget) in salinity at high pressure and the deviations you give in Figure 12?

**Authors**: We are not quite sure what exactly is being asked. Therefore, we try to give step-by-step feedback.

**Reviewer**: 'At high pressure, the linearity was checked and corrected with 4 substances, the less precise having the higher densities.'

**Authors**: Certainly our publication describing the development and validation of the substitution densimeter[4] is referred to.

*Linearity check.* In order to check the linearity of the substitution densimeter for high pressure, we performed comparison measurements against our substitution densimeter for atmospheric pressure that had been checked against a hydrostatic weighing densimeter before. To this end, we had premixed salt solutions and measured samples therefrom in each apparatus. Thus, the weighing densimeter served as reference to the atmospheric-pressure densimeter that again served as reference to the high-pressure densimeter. This linearity checks were performed at atmospheric pressure. The agreement between all those comparison measurements was better than $\pm 2 \, \mathrm{g \, m^{-3}}$, even for the high pressure densimeter.

*Adjustment.* The adjustment of the high-pressure densimeter is performed in two stages.

In the first stage the densimeter is adjusted for atmospheric pressure. Here, three liquids were used: *n*-nonane, water and tetrachloroethylene, each having densities at $20 \, \mathrm{°C}$ of $720 \, \mathrm{kg \, m^{-3}}$, $998 \, \mathrm{kg \, m^{-3}}$, and $1622 \, \mathrm{kg \, m^{-3}}$. The standard uncertainty in tetrachloroethylene reference densities was $25 \, \mathrm{g \, m^{-3}}$.

In the second stage the densimeter is adjusted for high pressure. Here, only water was used.

**Reviewer**: 'In this publication, their standard uncertainty is given to be $25 \, \mathrm{g \, m^{-3}}$ [..]'

**Authors**: Do you mean the uncertainty in the tetrachloroethylene reference densities used in adjustment for atmospheric pressure?

**Reviewer**: '[..] and you give a standard uncertainty in adjustment of $19 \, \mathrm{g \, m^{-3}}$ at $65 \, \mathrm{MPa}$.'

**Authors**: Certainly the uncertainty budget for a substitution measurement of seawater at $65 \, \mathrm{MPa}$ is referred to, i.e. Table 5 [4]. This budget is based on the uncertainty model described in Appendix A [4]. The standard uncertainty in adjustment of $19 \, \mathrm{g \, m^{-3}}$ that is given in this table can only be used, if the correlation between the water and seawater density in the substitution measurement is taken into account.

**Reviewer**: 'In this case, how can you ensure an uncertainty of 0.008 (including the linearity in the budget) in salinity at high pressure [...]'

**Authors**: The uncertainty in salinity of 0.008 corresponds to the uncertainty in relative density, i.e. relative to water, of $6 \, \mathrm{g \, m^{-3}}$. This uncertainty results from the uncertainty budget (of Table 5 [4]), which describes the uncertainty in absolute density, if the uncertainty of the water reference density is taken out.

**Reviewer**: '[..] and the deviations you give in Figure 12?'

**Authors**: The deviation in Figure 12 has to be evaluated against its uncertainty. The deviation is within the uncertainty, thus there is consistency for high pressure.
* * *
[4] Schmidt, H., Wolf, H. and Hassel, E.: A method to measure the density accurately to the level of 10⁻⁶, Metrologia, 53, 2, 770–86, doi:10.1088/0026-1394/53/2/770, 2016.

Are there details on the substitution densimeter yet not provided?

**What is the impact of the in/output assembly on the substitution density measured?**

Measuring the density not using the substitution method: after a quick-adjustment, we never observed deviations $> \pm 5 \, \mathrm{g \, m^{-3}}$, even if the densimeter was filled manually using syringes that were directly and coarsely connected to the inlet of the oscillating U-tube. The inlet may be mechanically decoupled to avoid such deviations.

Measuring the density using the substitution method (and a permanent filling installation): such effects are eliminated, as the impact on the measurement of water and seawater density is identical and no parts are being moved during the measurements.

The density is determined from the oscillation frequency and has to be corrected for damping effects that are caused by the friction between fluid layers due to viscosity. To correct for this effect, the first harmonic oscillation frequency is used by the firmware[5]. The impact of the in/output assembly on density measurement therefore has to be considered before this background. Frankly speaking, the damping-corrected density has to be used instead of the non-damping-corrected density. Using the non-damping-corrected density in a substitution measurement of seawater can, based on our experience, cause the density being measured too high by up to $10 \, \mathrm{g \, m^{-3}}$, as any damping effects force the base frequency being too low, thereby pretending a higher density.

**The DMAs inclination affects the measurement density by $2 \, \mathrm{g \, m^{-3}}$ per $1 \degree$ (degree). How was this considered? (Q. 3.2)**

The effect is well-known, we measured exactly $1.82 \, \mathrm{g \, m^{-3}}$ per $1 \degree$. The correction is multiplicative as gravitational forces are working only partly, if the inclination is not perpendicular to the direction of the gravitational force. The effect is therefore eliminated by an adjustment at the same inclination.

Apart from the fact that our DMAs are set up on fixed straight surfaces, such deviations are corrected by the substitution method. If a DMA is used without the substitution method, it can also be quick-adjusted (with air and water) at an inclination and then used afterwards. The decisive factor is that the inclination does not change between adjustment and measurement. This may be a problem in measurements aboard a ship.

**How do you ensure that the tube is sufficiently long to avoid diffusion of the oil into the U-tube of the DMA HP during measurements at high pressure? (Q. 3.3)**

The oil and water are not miscible, so there is always a phase boundary between both liquids. The phase boundary is reinforced, as the capillary tube in that both liquids meet has a small diameter of 1.6 mm. The density of the oil is lower than that of the water. We took advantage of that and installed the density measurement part at a lower level than the pressurization part that is filled with oil, thereby preventing convection downwards (into the DMA).

In addition, the filling line was rinsed with ethanol and water after each substitution measurement, thereby removing any oil that may stick to and creep on the inner tube wall, as such oil remains can disturb a clean replacement of seawater by water and vice versa. If, nonetheless, there is such a disturbance, this is seen in an increase in the water density measured or decrease in the seawater density measured, provided a measurement series (water – seawater – water – seawater – … – water) is conducted. For the high pressure measurements, we performed usually 5 repeated substitution measurements per temperature-pressure density point, i.e. 11 liquid
* * *
[5] Stabinger, H.: Density measurement using modern oscillating transducers, South Yorkshire Trading Unit, Sheffield, 1994.

replacements (water was always first and last); following the cleaning routine an impure replacement of liquids never occurred.

Page 9 Line 15: Replace 'to store the of seawater' by 'to store the seawater'. (H. 4)

The following change was made to the manuscript (from left to right):

| The borosilicate vessels used to store the  seawater samples [..] | The borosilicate vessels used to store the seawater samples [..] |
|---|---|

**Why do we speak of a linear dependence (on salinity) of the deviations of the Millero et al.-measurements from the density–salinity relation, when the deviations (shown in Fig. 11b) may suggest another type of dependence, too, e.g. a quadratic? (Q. 5)**

The linear dependence (on salinity) of the density deviation between TEOS-10 and the density–salinity relation is obvious. To develop TEOS-10 in this salinity-temperature region (at atmospheric pressure) a combined dataset[6] that consists of density data of Millero et al.[7] and Poisson et al.[8] was used[9,10]. However, for combining, both the data were adjusted. In Fig. 11b (of the discussion paper) the adjusted data is shown.

In order to manifest the linear dependence, we should perhaps use the original magnetic float data of Millero et al.[7] that was converted to ITS-90 to show its deviations to the density–salinity relation in the left figure below. In this figure, the magnetic float data is separated into density measurements that were explicitly performed in a closed cell[6], thereby preventing evaporation of water during measurement, and into those that were putatively performed in an open cell, thereby allowing evaporation. If water evaporates from the seawater, depending on the duration and the sequence of measurements, the salinity and density increase. The densities for $S < 30$ are therefore systematically too high, which explains the upward scatter in deviations for $S < 30$ compared to that for $S \geq 30$. Thus, only the data with the smaller deviations, i.e. the lower points in left figure, in this salinity range should be considered to investigate the dependence on salinity. The right figure shows the deviations of the left figure normalized using the mean deviation of the measurements made with a closed cell (without evaporation). The smaller deviations for $S < 30$ highly correlate with the zero-line suggesting a linear dependence on salinity.

Note that the density data for $S = 40$ is not shown, as these measurements may be affected from the preparation by evaporation; actually, these densities, i.e. the salinities, were found to be too high, when combining the density data of Millero et al. and Poisson et al.[6]
* * *
[6] JPOTS: Background papers and supporting data on the international equation of state of seawater 1980, Unesco Division of Marine Sciences, web: http://unesdoc.unesco.org/images/0004/000473/047363eb.pdf, 1980.

[7] Millero, F. J., Gonzales, A. and Ward, G. K.: The density of seawater solutions at one atmosphere as a function of temperature and salinity, Journal of Marine Research, 34, 61–93, 1976.

[8] Poisson, A., Brunet, C. and Brun-Cottan, J. C.: Density of standard seawater solutions at atmospheric pressure, Deep-Sea Research, 27A, 1013–28, doi:10.1016/0198-0149(80)90062-X, 1980.

[9] Feistel, R.: A Gibbs function for seawater thermodynamics for -6 to 80°C and salinity up to 120 g/kg, Deep-Sea Research, 55, 1639–71, doi:10.1016/j.dsr.2008.07.004, 2008.

[10] Feistel, R.: A new extended Gibbs thermodynamic potential of seawater, Progress in Oceanography, 58, 43–114, doi:10.1016/S0079-6611(03)00088-0, 2003.

[Figure]

**Fig.** Deviation of densities obtained from standard seawater using a magnetic float densimeter by Millero et al.[6] from the density–salinity relation (M. et al. minus DSR). Absolute deviations (a) and salinity-35-normalized deviations (b).

---

## Referee Comment (RC4) · Anonymous Referee #2 · 29 Sep 2017

**Comments on the answers of the authors of the publication**

**"The density-salinity relation of standard seawater"**

The authors have completely answered to my first question about the effect of using the Anton Paar's formula for the calculation of the air density, instead of the BIPM one's. They have evaluated its impact and the impact of a zero-drift on the accuracy of seawater measurements. These points are not details, and even if they lead small uncertainties, according to me, they might be mentioned in the paper and included in the uncertainty budget to take off any doubt on the validity of their measurements.

About the substitution method, the authors have completed the paragraph 2.1.

Concerning the uncertainties at high pressure, the given explanations correspond to contain of the Metrologia's publication and they are clear, but the explanations on the relative density budget are less clear, so that the explanations of the paragraph 2.4. The calculation of seawater density relative to water $\rho_{mes}{}^{SW} - \rho_{mes}{}^{H2O}$ allows the subtraction of the linearity errors at $\rho_{mes}{}^{H2O}$ and $\rho_{mes}{}^{SW}$, and the reduction of the errors, but if the adjustment is made with an uncertainty of 19 $g/m^3$, this uncertainty stays the same.

Concerning the figure 12 it is OK for the uncertainties of 8, 26 and 33 $g/m^3$ given in the legend and my remark was unfounded.

About the apparatus: thank you for the details given on the in/output assembly, on the effects of inclination, and on how the diffusion of oil in the U-tube is avoided.

About the linear dependence on salinity, I understand the arguments given and I appreciate the figures joined to the explanations.

---

## Author Comment (AC4) · 2 Oct 2017

**Response to RC4 (29 Sep 2017)**

**General comments**

We appreciate the care taken in reviewing our work.

The impact of a densimeter zero-drift is more likely to be related to our previous publication on our substitution method and is negligible in case of regular densimeter maintenance. Therefore, in this publication, we suggest to provide such information in a reader-oriented way in the digital supplement and add a hint thereto in the method section.

**Specific comments**

**Reviewer**: Concerning the uncertainties at high pressure, the given explanations correspond to contain of the Metrologia's publication and they are clear, but the explanations on the relative density budget are less clear, so that the explanations of the paragraph 2.4. The calculation of seawater density relative to water, $\rho_{\text{mes}}^{\text{SW}} - \rho_{\text{mes}}^{\text{H}_2\text{O}}$, allows the subtraction of the linearity errors at $\rho_{\text{mes}}^{\text{SW}}$ and $\rho_{\text{mes}}^{\text{H}_2\text{O}}$, and the reduction of the errors, but if the adjustment is made with an uncertainty of $19 \text{ g m}^{-3}$, this uncertainty stays the same.

**Authors**: We are still not quite sure what exactly is being asked, therefore we give step-by-step feedback.

**Reviewer**: [..] but the explanations on the relative density budget are less clear, so that the explanations of the paragraph 2.4.

**Authors**: Are the explanations of the uncertainty budget in our previous publication referred to?

**Reviewer**: The calculation of seawater density relative to water, $\rho_{\text{mes}}^{\text{SW}} - \rho_{\text{mes}}^{\text{H}_2\text{O}}$, allows the subtraction of the linearity errors at $\rho_{\text{mes}}^{\text{SW}}$ and $\rho_{\text{mes}}^{\text{H}_2\text{O}}$, and the reduction of the errors, but if the adjustment is made with an uncertainty of $19 \text{ g m}^{-3}$, this uncertainty stays the same.

**Authors**: The uncertainty contribution from adjustment, more precisely that of the adjustment reference densities, to a relative density measurement on seawater with salinity 35 at 65 MPa is $\approx 9 \text{ g m}^{-3}$, which is less than $19 \text{ g m}^{-3}$, as the correlation coefficient has to be considered. Additionally, the uncertainty contribution of oscillating U-tube dimensions and material properties necessary for the adjustment model are considered by varying these quantities in the final adjustment equation. The uncertainty was therefore estimated conservatively. The uncertainty contributions are included in the relative density uncertainty given in the digital supplement and actually stay the same.

The uncertainty in the density–salinity relation was estimated as described in Appendix B. This suggested a maximum uncertainty of $6 \text{ g m}^{-3}$ in the interpolation region. This is mainly a result of the very good ability to fit the data, as seen in Fig. 8 – although the relative density uncertainty is considered in the fitting process, the residuals are almost all within $6 \text{ g m}^{-3}$. This again may be a result of the conservative estimation of uncertainty, i.e. an overestimation of the uncertainty.

It should be noted that the positive deviation of TEOS-10 from the density–salinity relation for high pressures seen in Fig. 12 is significantly reduced, if the deviation found for atmospheric pressure is taken out.

---

## Referee Comment (RC5) · Anonymous Referee #2 · 6 Oct 2017

I regret that the explanations on the calibration of the instrument to the air density doesn't be integrated in the paper, but if a hint is added in the publication and that the digital supplement is easily accessible, I am satisfied.

Response to the specific comments of the authors: Question 1 : Are the explanation of the uncertainty budget in our previous publication referred to? Yes. The explanations given in the following of the response are sufficient to clear up the doubts I had on the values given in the document.

I close the discussion and I accept the publication of the paper.